# XCO2 estimates from the OCO-2 measurements using a neural network approach

5  Leslie David, François-Marie Bréon, Frédéric Chevallier

Laboratoire des Sciences du Climat et de l'Environnement/IPSL,
CEA-CNRS-UVSQ, Université Paris-Saclay, F-91198 Gif-sur-Yvette, France
*Correspondence to*: Francois-Marie Breon (fmbreon@cea.fr)

10  **Abstract**. The OCO-2 instrument measures high-resolution spectra of the sun radiance reflected at the Earth surface or scattered in the atmosphere. These spectra are used to estimate the column-averaged dry air mole fraction of CO2 (XCO2) and the surface pressure. The official retrieval algorithm (NASA's Atmospheric CO2 Observations from Space retrievals - ACOS) is a *full physics algorithm* and has been extensively evaluated. Here we propose an alternative approach based on an artificial neural network (NN) technique. For the training and evaluation, we use as reference estimate (i) the surface pressures from a numerical weather model and (ii) the XCO2 derived from an atmospheric transport simulation constrained by surface air-sample measurements of CO2. The NN is trained here using real measurements acquired in nadir mode on cloud-free scenes during even months and is then evaluated against similar observations of odd months. The evaluation indicates that the NN retrieves the surface pressure with a root-mean-square error better than 3 hPa and XCO2 with a 1-sigma precision of 0.8 ppm. The statistics indicate that the NN, that has been trained with a representative set of data, allows excellent accuracy, slightly better than that of the full physics algorithm. An evaluation against reference spectrophotometer XCO2 retrievals indicates similar accuracy for the NN and ACOS estimates, with a skill that varies among the various stations. The NN-model differences show spatio-temporal structures that indicate a potential for improving our knowledge of CO2 fluxes. We finally discuss the pros and cons of using this NN approach for the processing of the data from OCO-2 or other space missions.

25  **1.  Introduction**

During the past decades, natural fluxes have absorbed about half of the anthropogenic emissions of CO2 (Knorr, 2009), but there is large uncertainty on the spatial distribution of this sink over time and therefore on the processes that control it. A growing network of high-precision atmospheric CO2 measurements has been used together with meteorological information to constrain the sources and sinks of CO2 using a technique known as *atmospheric inversion* (e.g., Peylin et al., 2013), but the lack of data in large regions of the globe like the tropics does not allow monitoring these fluxes with enough space-time resolution. Early attempts to complement this network with satellite retrievals from sensors that were not specifically designed for this purpose were not successful (Chevallier et al., 2005), but a series of dedicated instruments have been put in orbit since the Greenhouse Gases Observing Satellite (GOSAT, Yokota et al., 2009) and the second Orbiting Carbon Observatory (OCO-2 Eldering et al., 2017a), launched in 2009 and 2014, respectively, and still operated at the time of writing. This new and evolving constellation is directly supported by Japanese, US, Chinese and European space agencies (CEOS Atmospheric Composition Virtual Constellation Greenhouse Gas Team, 2018). All missions have adopted the same CO2 observation principle that consists in measuring the solar irradiance that has been

reflected at the Earth's surface in selected spectral bands. Along the double atmospheric path (down-going and up-going), the sunlight is absorbed by atmospheric molecules at specific wavelengths. The resulting absorption lines on the measured spectra makes it possible to estimate the amount of gas between the surface and the top of the atmosphere. $CO_2$ shows many such absorption lines around 1.61 and 2.06 μm that are used to estimate the $CO_2$ column. Similarly, the oxygen lines around 0.76 μm are used to estimate the surface pressure and can also be used to infer the sunlight atmospheric path, leading to the column-averaged dry air mole fraction of $CO_2$, referred to as XCO2 (O'Brien and Rayner, 2002, Crisp et al., 2004).

One main difficulty in the retrieval of XCO2 from the measured spectra results from the presence of atmospheric particles that scatter light and change its atmospheric path. Accounting for aerosols, in particular, is challenging because aerosols are very variable in amount and in vertical distribution. Another major difficulty results from modelling errors. The radiative transfer models that are used for the retrieval leave significant residuals between the measured and modelled spectra, even after the XCO2 and aerosol amount have been inverted for a best fit (Crisp et al., 2012; O'Dell et al., 2018).

As a consequence of the various uncertainties in the retrieval process, raw XCO2 retrievals show significant biases against reference ground-based retrievals (Wunch et al., 2011b, 2017). These biases, together with the comparison against modelling results, led to the development of empirical corrections to the retrieved XCO2. In the case of the OCO-2 v8r retrievals generated by NASA's Atmospheric $CO_2$ Observations from Space (ACOS), these corrections amount to roughly half that of the "signal", i.e. of the difference between the prior and the retrieved XCO2 (O'Dell et al. 2018).

The limitations in the full-physics retrieval method, despite considerable efforts and progresses (e.g., O'Dell et al. 2018, Reuter et al. 2017, Wu et al., 2018 in the case of OCO-2), encourage developing alternative approaches. Here, we want to re-evaluate the potential of an artificial neural network technique (NN) to estimate XCO2 from the measured spectra. A NN-based technique was already used by Chédin et al. (2003) for a fast retrieval of mid-tropospheric-mean $CO_2$ concentrations from some meteorological satellite radiometers. These authors trained their NNs on a large ensemble of radiance simulations made by a reference radiation model and assuming diverse atmospheric and surface conditions. NN-based approaches are also commonly used for the retrieval of other species from various high-spectral-resolution satellite radiance measurements because of their computational efficiency (e.g., Hadji-Lazaro et al. 1999).

A NN approach requires a large and representative training dataset. A standard method for problems similar to that discussed here is to use a radiative transfer model and to generate a large ensemble of pseudo observations based on assumed atmospheric and surface parameters. However, as mentioned above, the radiative transfer models have deficiencies that are rather small, but nevertheless significant with respect to the high precision objective of the $CO_2$ measurements. In addition, there may be some wrong assumptions and unknown instrumental defects that are not accounted for in the forward modeling. We thus prefer to avoid using such radiative transfer models and rather base the training on a fully empirical approach (see, e.g., Aires et al., 2005). We use real OCO-2 observations together with collocated estimates of the surface pressure and XCO2. The retrievals from the NN approach are evaluated against model estimates of surface pressure and XCO2, as well as observations from the Total Carbon Column Observing Network (TCCON, Wunch et al., 2011). In the following, section 2 presents the approach while section 3 describes the results. Section 4 discusses the results and the way forward.

## 2. Data and method

Our NN estimates XCO2 and the surface pressure from nadir spectra measured by the OCO-2 satellite over land. OCO-2 has eight cross-track footprints (e.g., Eldering et al., 2017), but we only use footprint #4 in the following for simplicity.

If successful, the same approach can be applied to all footprints. The focus on nadir measurements here is motivated by the complication introduced by the Doppler effect in glint mode, which is the other pointing mode for OCO-2 routine science operations: the absorption lines affect pixel elements that vary among the spectra. These variations of the position of the absorption line may cause additional difficulty to the NN training. The solar lines in the nadir spectra are also affected by Doppler shifts due to the motion of the Earth and satellite relative to the sun, but this concerns a limited set of spectral elements that are affected by the solar (Fraunhofer) lines. The development of a glint-mode NN is therefore left for a future study.

We use spectral samples in the three bands of the instrument (around 0.76, 1.61 and 2.06 μm). They have footprints of ~ 3 km$^2$ on the ground. In principle, each band is described by 1016 pixel elements but some are marked as bad either because some of the corresponding detectors died at some stage or because of known temporary or permanent issues. We systematically remove 15 pixel elements that are flagged in about 80% of the spectra and 478 pixels in the band edges. Conversely, we do not remove the spectra that are affected by the deep solar lines, and we let the NN handle these specific features. Because the information in the spectrum is mostly in the relative depth of the absorption lines, and not in their overall amplitude, we normalize each spectrum by a radiance that is representative of the offline values (i.e. the mean of the 90-95% range for each spectrum). This essentially removes the impact of the variations in the surface albedo and in the sun irradiance linked to the solar zenith angle. Other choices for the input may be attempted in the future.

As input to the NN, we add the observation geometry (sun zenith angle and relative azimuth). The sun zenith angle drives the atmospheric pathlength and is then required for the interpretation of the absorption line depth in terms of atmospheric optical depth. The azimuth was not included in our first attempts but, when included, it led to a significant improvement in the results. Although the NN technique does not allow for a clear physical interpretation, we assume that the information brought by the relative azimuth is linked to the polarization of the molecular scattering contribution to the measurements that varies with the azimuth.

The NN exploits these 2557 input variables to compute 2 variables only: XCO2 and the surface pressure. It is structured as a Multilayer Perceptron (Rumelhart et al. 1986) with one hidden layer of 500 neurons that use a sigmoid activation function. The number of hidden layers is somewhat arbitrary and based on a limited sample of trials. Lower quality estimates were obtained with 50 neurons whereas the training time increased markedly for 1000 neurons and more. The weights of the input variables to the hidden neurons and the weights of the hidden variables to the output variables are adjusted iteratively with the standard Keras library (Chollet, 2015). Figure A1 in the appendix illustrates the convergence process. The NN cost function (aka loss) becomes fairly constant for a test dataset after about 100 iterations, while it continues to decrease for the training dataset, indicating an over-fitting of the data. The iteration is stopped when there is no decrease of the test loss for 50 iterations. There is a factor of 3 to 4 between the loss of the training dataset and that of the test, which confirms the over-fitting of the former.

Note that the NN estimate does not use any a priori information on surface pressure or the CO2 profile after the training is done. Also, no explicit information is provided on the altitude, location or time period of the observation. The NN estimates are therefore only driven by the OCO-2 spectrum measurements, together with the observation geometry (sun zenith and relative azimuth). The observation geometry varies with the latitude and the season so that the NN may infer some location information from this input. Conversely, it is the same from one year to the next and, at a given date, for all longitudes. Thus, there is no information on the longitude or the year of observation in the geometry parameters that are provided to the network.

The NN training is based on OCO-2 radiance measurements (v8r) acquired during even months between January 2015 and August 2018. The 4-year period allows varying the global background CO2 dry air mole fraction by ~ 2%, as much as typical XCO2 seasonal variations in the northern extra-tropics (see, e.g., Fig. 1 of Agustí-Panareda et al., 2019). Our evaluation dataset is based on observations during the odd months of the same period. In both cases, we make use of
XCO2 estimates and the quality control filters of the ACOS L2Lite v9r products: only observations with *xco2_quality_flag*=0 are used. We also consider the *warn level*, *outcome flag* and *cloud_flag_idp* that are provided in the v8r L2lite and L2Std files. For the training of the NN, we only use the best quality observations, i.e. those with a *warn level* lower or equal to 2, a *cloud_flag* of 3 (very clear) and an *outcome flag* of 1. This choice is based on an evaluation of the surface pressure estimates that is described below (with the description of Figure 3). This distinction leads to about
131 000 observations for the training. For the evaluation of the NN estimates, we use less restrictive criteria and accept observations with outcome_flag of either 1 and 2, and cloud_flag of 2 or 3. These choices are justified below. The spatial distribution of the observations that are used for the training is shown in Figure A2 of the appendix. The training dataset covers most regions of the globe with the exception of South America. The underrepresentation of this sub-continent stems for both the high cloudiness and impact of cosmic rays that leads to missing pixel elements (see below).

For the reference surface pressure (training and evaluation), an obvious choice is the use of numerical weather analyses corrected for the sounding altitude. Indeed, the typical accuracy for surface pressure data is on the order of 1 hPa (Salstein et al. 2008). For convenience, we use the surface pressure that is provided together with the OCO-2 data and that is derived from the Goddard Earth Observing System, Version 5, Forward Processing for Instrument Teams (GEOS5-FP-IT) created at Goddard Space Flight Center Global Modeling and Assimilation Office (Suarez et al. 2008 and Lucchesi
et al. 2013). There is no such obvious choice for XCO2 as there is no global-scale highly-accurate dataset of XCO2 and we thus rely here on best estimates from a modelling approach. We use the CO2 atmospheric inversion of the Copernicus Atmosphere Monitoring Service (CAMS, atmosphere.copernicus.eu, last access: 28 January 2020, Chevallier et al., 2010); version 18r2). This product was released in July 2019 and contributed, e.g., to the Global Carbon Budget 2019 of Friedlingstein et al. (2019). It results from the assimilation of CO2 surface air-sample measurements in a global
atmospheric transport model run at spatial resolution 1.90° in latitude and 3.75° in longitude over the period 1979-2018 and using the adjoint of this transport model. Neither satellite retrievals nor TCCON observations were used for this modelling. For each OCO-2 observation, XCO2 is computed from the collocated concentration vertical profile, through a simple integration weighted by the pressure width of the model layers. Note that the model layers use "dry" pressure coordinates so that there is no need for a water vapor correction in the vertical integration. The GEOS5-FP-IT surface
pressure and the XCO2 from CAMS are used both for the training and the evaluation, although using independent datasets (odd and even months).

Many measured spectra lack one or several spectral pixels. This is particularly the case over South America, as a consequence of the South Atlantic cosmic ray flux anomaly that impacts the OCO-2 detector in this region. We therefore devised a method to interpolate the spectra and to fill the missing pixels. Our method first sorts all spectral pixels as a
function of the measured radiance in a large number of complete measured spectra. The pixel ranks are averaged to generate a rank representative of the full dataset. Then, when a pixel element is missing in a spectrum, we look for its typical rank and we average the radiances of the two pixel elements that have the ranks just above and below. The procedure is applied even when several pixel elements are missing in a spectrum, except when these are successive in the typical ranking. The procedure described here fills the missing elements, and the NN can then be applied to the corrected
spectrum to estimate the surface pressure and XCO2.

## 3. Results

Figure 1 shows a density histogram of the GEOS5 FP-IT surface pressure analysis and of the NN estimate for the evaluation dataset (odd months). Clearly, there is an excellent agreement between the two over a very wide range of surface pressures. There is no significant bias and the standard deviation is 2.9 hPa. The equivalent ACOS v8r retrieval shows a bias of 1.5 hPa and a standard deviation of 3.4 hPa, slightly larger than that of the NN approach. Note that the ACOS statistics are those of the ACOS retrieval-minus-prior statistics (see Section 2). Interpreting them in terms of error is counter-intuitive because the Bayesian retrieval is supposed to be better than the prior, but in practice radiation modelling errors lead to a different interpretation (see, e.g., the discussion in Section 4.3.4 of O'Dell et al. 2018).

Both NN and ACOS correlations with GEOS5 FP-IT are very high (0.997 and 0.996) although the best fit shows a very small deviation from the 1:1 line. Interestingly, the best fit deviations from the 1:1 line are of opposite sign (slopes 0.99 and 1.01). The results of the NN are surprisingly good given the simplicity of the approach and given that the NN estimate does not use any a priori information or ancillary information such as the surface altitude or temperature profile, contrarily to the ACOS estimate. The quality of the NN results for the estimate of the surface pressure is a first demonstration of the potential of the approach. Note that the retrieval accuracy holds over a very large range of surface pressures (the relative variations of XCO2 are much smaller), although there is some indication of biases for the lowest pressures that are under-represented in the training dataset. These biases of ≈5 hPa affect the observations over high elevation surfaces such as the Tibetan Plateau or the US Rocky Mountains.

Figure 2 is similar to Figure 1 but for XCO2. There is no significant bias between the NN estimate and the CAMS model, while the standard deviation is 0.84 ppm. The bias-corrected ACOS retrievals show a slight bias against the CAMS model and the standard deviation (1.14 ppm) is larger than that of the NN approach. Note that the statistics given here are affected by CAMS modelling errors that may eventually be corrected with the help of the satellite information. The best fit slope deviations from the 1:1 line are larger than for the surface pressure: the slopes are 0.93 for the NN and 0.87 for ACOS.

Figure 1 and 2, together with the quantitative assessment of the precision are given for the observations that are clear according to ACOS (Cloud Flag=2 or 3), that have a warn level of 2 or less, that may include missing pixel elements, and that have an outcome flag of 1 or 2. This choice is based on a prior performance analysis. We have analyzed how the performance of the NN approach varies with the quality indicators. For this objective, we have compared the retrieved surface pressure against the value derived from the numerical weather data, as in Figure 1, and we have evaluated the statistic of their difference as a function of the quality flags. First (figure not shown), there is no significant difference between the cases when the measured spectra are complete and those when one or several missing pixel elements have been interpolated with the method described above. Conversely, the statistics vary with the cloud flag and the warn level, as shown in Figure 3. We only use the spectra for which an ACOS retrieval is available. Among those, and according to the flag *cloud_flag_idp*, about 53% are labeled as "very clear" while 43% are "probably clear". The statistics are slightly better for the former than they are for the latter. Conversely, the rather rare "definitely cloudy" and "probably cloudy" show deviations that are significantly larger. This result was well expected since our NN did not learn how to handle clouds in the spectra, so that all "definitely cloudy" and "probably cloudy" soundings are outside the domain covered by the training dataset. Note also that the observations used here have all been classified as "clear" by the ACOS pre-processing. Thus, most OCO-2 observations are not used here and Figure 3 should not be interpreted as the ability to retrieve the surface pressure in cloudy conditions. Most (78%) of the observations have a warn level of 0. The deviation statistics increase with the warn level, both in terms of bias and standard deviation. In comparison, the difference in the

statistics for an outcome flag of 1 and 2 are small. Besides, more than half of the ACOS retrievals have an outcome flag of 2 which encourages us not to reject those for further use. Based on this analysis, we retain all spectra that are very clear (cloud flag of 2 or 3) and that have a warn level of 2 or less.

We have made a similar figure as Figure 3 but based on the XCO2 estimates (not shown). Although the results are similar in terms of sign (i.e. increase of the deviations with the warn levels), the signal is not as obvious (there is less relative difference between a warn level and another, or for the various cloud flags). Our interpretation is that the relative accuracy of the surface pressure that is used as a reference estimate is much better than that of the NN retrieval, whereas the accuracy of the XCO2 from CAMS is not much better than that of the NN. As a consequence, variations in the accuracy of the NN do not show up as clearly for XCO2 than they do for the surface pressure.

A standard method to evaluate an algorithm that estimates XCO2 from spaceborne observation is the comparison of its products against estimates from TCCON retrievals. These estimates use ground-based solar absorption spectra recorded by Fourier transform infrared spectroscopy and have been tuned with airborne in-situ profiles (Wunch et al. 2010). To take advantage of the full potential of the TCCON retrievals for the bias-correction and validation of the XCO2 estimates, the OCO-2 platform can be oriented so that the instrument field of view is close to the surface station. The ACOS full-physics algorithm can handle these spectra that are acquired in neither nadir nor glint geometries, but the NN was trained solely on nadir spectra and cannot be applied yet to the observations acquired in target mode. We thus have to rely on nadir measurements acquired in the vicinity of TCCON sites. In the following, we use nadir measurements that are within 5 degrees in longitude and 1.5 degrees in latitude to the TCCON site. The XCO2 estimates (either from ACOS, the NN, or the model sampled at the OCO-2 measurement location) are averaged for a given overpass. Similarly, we average the TCCON estimates of XCO2 within 30 minutes of the satellite overpass. No attempt was made to correct for the different weighting functions of the surface and spaceborne remote sensing estimates. The comparisons are shown in Figure 4 for each TCCON station of Table 1.

Overall, the biases and standard deviations of the differences to TCCON observations are -0.34±1.40 ppm for the NN, -0.47±1.49 ppm for ACOS and 0.04±1.09 ppm for CAMS. Statistics per stations are provided in Table 1. Two stations, Paris and Pasadena, show a large negative bias for both estimates, which may be interpreted as the impact of the city on the atmosphere sampled by the TCCON measurement, while the atmosphere sampled by the distant satellite may be less affected. Conversely, there is no such negative bias for other stations that are located close to large cities, such as Tsukuba that is in the suburb of the Tokyo Metropolitan area. Zugspitze is rather specific due to its high altitude. The comparison against TCCON indicates that the NN approach has a similar performance as ACOS, if not better. The dispersion is larger for one versus the other for some stations, while the opposite is true for others. Note also that the CAMS model performs better than both satellite retrievals for most stations. This observation provides further justification to the use of this model for training the NN.

The evaluation of the algorithm performance is limited by the distance between the satellite estimate and its surface validation. This is inherent to the use of nadir-only observations that are seldom located close to the TCCON sites. A reduction of the distance results in less coincidences, which leads to a validation dataset of poor representativeness. Note that the CAMS model was sampled at the location of the satellite observations, so that the higher performance of the model versus the satellite products cannot be caused by a higher proximity to the TCCON station.

We now investigate whether the model-minus-NN differences are purely random or contain some spatial or temporal structures. This question is important as, if the differences show a random structure, there is little hope to use these data to improve the surface fluxes used in the CAMS product. Conversely, if the XCO2 differences do show some structures,

they can be attributed to surface flux errors in the CAMS product that may then be corrected through inverse atmospheric modelling. There is no certainty, however, as a spatial structure in the NN-minus-CAMS difference can also be interpreted as a bias in the satellite estimate.

We first show (Figure 5) the difference between the NN estimate of the surface pressure and that from the numerical weather analyses. These are monthly maps of the NN-minus-CAMS difference for the 3 years of the period at a 5°×5° resolution. We only present the odd months as the others months have been used for the training, and therefore do not show any significant differences. There are very clear spatial patterns of a few hPa which are not expected and should be interpreted as a bias in the NN approach. The biases over the high mountains and plateaus have already been mentioned. In addition, positive biases tend to occur in the high latitudes, and negative biases toward the tropics. The structures show additional spatial and temporal patterns and are therefore more complex than just a latitude function. The same figure but based on the ACOS retrievals (Figure A3) displays large-scale structures with different spatial patterns: the surface pressure bias is mostly negative over the Northern latitudes and positive over the low latitudes. A histogram (Figure 6) of the monthly differences such as those shown on Figure 5 confirms that the amplitude of the surface pressure biases is larger with ACOS than it is with the NN. The NN (resp. ACOS) surface pressure bias is -0.33 hPa (resp 1.39 hPa) and the standard deviation is 2.12 (resp. 2.79 hPa).

Figure 7 is similar as Figure 5 but for XCO2 difference between the NN estimate and the CAMS model. As for the surface pressure, there are clear spatial patterns, with amplitudes of 1 to 2 ppm. The question is whether these are mostly linked to monthly biases in the CAMS model or to the NN. The first hypothesis is of course more favorable as it would indicate that the satellite data can bring new information to constrain the surface fluxes. However, the analysis of the surface pressure that shows biases of several hPa suggests that the NN XCO2 estimate may also show biases with spatially coherent patterns. Interestingly, the patterns vary in time and are not correlated with those of the surface pressure. Further analysis, in particular atmospheric flux inversion, is necessary for a proper interpretation of the NN-CAMS differences.

The differences of ACOS estimates to the CAMS model also show patterns of similar amplitude as those in Figure 7 (Figure A4). However, there is no clear correspondence between these patterns and those obtained using the NN product. The differences between the satellite products and the CAMS model are small, but these contain the information that may be used to improve our knowledge on the surface fluxes. The absence of a clear correlation between the spatio-temporal pattern from the NN and ACOS approaches indicate that their use would lead to very different corrections on the surface fluxes, if used as input of an atmospheric inversion approach. Figure 6, top, shows the histogram of these monthly-mean differences. The histograms are very similar for the two satellite products, although the standard deviation of the difference to the CAMS model is slightly larger for ACOS than it is for the NN approach (0.89 vs 0.83 ppm).

## 4. Discussion and Conclusion

The use of the same product for the NN training and its evaluation may be seen as a weakness of our analysis. One may argue that the NN has learned from the model and generates an estimate (either the surface pressure or XCO2) that is not based on the spectra but rather on some prior information. Let us recall that the NN input does not contain any information on the location or date of the observation. This is a strong indication that the information is derived from the spectra as the NN does not "know" the CAMS value that corresponds to the observation location. Yet, the NN input also includes the observation geometry (sun angle and azimuth) that is somewhat correlated with the latitude and day-in-the-year. One may then argue that the NN learns from this indirect information on the observation location and then generates an estimate that is based on the corresponding CAMS value. However, since the observation geometry is exactly the same

from one year to the next, there is no information, direct or indirect, on the observation year in the NN input. Thus, the XCO2 growth rate, that is accurately retrieved by the NN method (see Fig. 7), is necessarily derived from the spectra. A similar argument can be made on the spatial variation across the longitudes.

To further demonstrate that the NN retrieves XCO2 from the spectra rather than from the prior, we made an additional experiment. The training is based only on even months. As a consequence, the prior does not include any direct information on the odd months. For the odd months, the best prior estimate here is a linear interpolation between the two adjacent even months. We can then analyze how the NN estimate compares with the CAMS product, that accounts for the true synoptic variability, and a degraded version of CAMS that is based on a linear interpolation between the two adjacent months. This comparison is shown in Figure 8. The center figure compares the true CAMS value and that derived from the temporal interpolation. As expected, both are highly correlated (the seasonal cycle and the growth rate are kept in the interpolated values) but show nevertheless a difference standard deviation of 0.89 ppm. This value can be interpreted as the synoptic variability of XCO2 that is present in CAMS but is not captured in the interpolated product. The comparison of the NN estimate against CAMS (right) and the interpolated CAMS (left) shows significantly better agreement to the former. Thus, the NN product does reproduce some XCO2 variability that is not contained in the training prior. It provides further demonstration that the NN estimates relies on the spectra rather than on the time/space variations of the training dataset.

The results shown above indicate that the NN approach allows an estimate of surface pressure and XCO2 with a precision that is similar or better than that of the operational ACOS algorithm. The lack of independent "truth" data does not allow a full quantification of the product precision and accuracy. However, there are indications that the accuracy on the surface pressure is better than 3 hPa RMS, while the precision (standard deviation) of XCO2 is better than 0.9 ppm. The data used for the XCO2 product evaluation has its own error that is difficult to disentangle from that of the estimate based on the satellite observation. It may also contain a bias that is propagated to the NN through its training.

One obvious advantage of the NN approach is the speed of the computation that is several orders of magnitude higher than that of the full physics algorithm. This is significant given the current re-processing time of the OCO-2 dataset despite the considerable computing power that is made available for the mission. It also bears interesting prospects for future XCO2 imaging missions that will bring even higher data volume (e.g., Pinty et al., 2017).

Another advantage is that the NN approach described in this paper does not require the extensive de-bias procedure which is necessary for the ACOS product (O'Dell et al 2018, Kiel et al. 2019). Per construction, there is no bias between the NN estimates and the dataset that is used for the NN training. The NN approach requires therefore less effort and manpower.

There are however a number of drawbacks for the NN approach that is described in this paper.

One obvious drawback is the use of a CO2 model simulation in the training while the main purpose of the satellite observation is to improve our current knowledge on atmospheric CO2 and its surface fluxes. Our argument is that, although the CAMS simulation used here has high skill (as demonstrated in Figure 4), it may have positive or negative XCO2 biases for some months and some areas. These biases are independent from the measured spectra so that the NN training will aim at average values. As a consequence, the NN product could in principle be of higher quality than the CAMS product, even though the same model has been used as the reference estimate for the training (see, e.g., Aires et al., 2005).

Another drawback of the NN approach is that it does not directly provide its averaging kernel. The averaging kernel vector reports the sensitivity of the retrieved total column to changes in the concentration profile (Connor et al., 1994). It is a combination of physical information (about radiative transfer) and of statistical information (about the prior information). It is needed for a proper comparison with 3D atmospheric models (e.g., Chevallier 2015). When comparing with model simulations, for instance for atmospheric inversion, we may wish to neglect the NN implicit prior information: this hypothesis leads to a homogeneous pressure weighting over the vertical, as this is the product that the NN was trained to simulate. Alternatively, we could decide to neglect the difference in prior information between the NN and the full physics algorithm and use typical averaging kernels of the latter. A third, more involving, option would be to perform a detailed sensitivity study of the NN, based on radiative transfer simulations.

Similarly, the current version of our neural network does not provide a posterior uncertainty. A Monte Carlo approach using various training datasets could be use in the future for such an estimate.

Also, the NN that was developed cannot be safely used to process observations that are acquired later than a few weeks after the last data of the training dataset, in order to keep the application within the variability range of the training data and despite the $CO_2$ growth rate. Therefore, the use of the neural network approach for near real time applications would require frequent updates of the training phase.

We acknowledge the fact that the NN product that is evaluated here is not fully independent from the ACOS product. Indeed, we use the cloud flag and the quality diagnostic from ACOS to select the spectra that are of sufficient quality. If we aim at some kind of operational product, there is a need to design a procedure to identify these good quality spectra. One option would be to compare the surface pressure retrieved by the NN to the numerical weather analysis estimate, and to reject cases with significant deviations (e.g. differences larger than 3 hPa).

Despite these drawbacks, the results presented here do show that a neural network has a large potential for the estimate of XCO2 from satellite observations such as those of OCO-2, of the forthcoming MicroCarb (Pascal et al. 2017) or the CO2M constellation (Sierk et al. 2018) that aims at measuring anthropogenic emissions. It is rather amazing that a first attempt leads to trueness and precision numbers that are similar or better than those of the full physics algorithm. There are several ways for improvement: one is to provide the NN with some ancillary information such as the surface altitude or a proxy of the atmospheric temperature. Another one is to train the NN with model estimates (such as those of CAMS used here) but that have been better sampled for their assumed precision, for instance through a multi-model evaluation. Also, one could train the NN with observations acquired during a few days of each month, rather than the even months as done here, so that the evaluation dataset would provide a better evaluation of the seasonal cycle.

Our next objective is to attempt a similar NN approach but for the measurements that have been acquired in the glint mode. As explained above, the glint observations may be more difficult to reproduce by the NN than those acquired in the nadir mode. However, we have been very much surprised by the ability of the NN with the nadir data, and cannot exclude to be surprised again. Last, we shall analyze the spatial structure of the NN retrievals in regions that are expected to be homogeneous and in regions where structures of anthropogenic origin are expected (e.g., Nassar et al., 2017; Reuter et al., 2019).

### Acknowledgments

This work was in part funded by CNES, the French space agency, in the context of the preparation for the MicroCarb mission. OCO-2 L1 and L2 data were produced by the OCO-2 project at the Jet Propulsion Laboratory, California Institute

of Technology, and obtained from the ACOS/OCO-2 data archive maintained at the NASA Goddard Earth Science Data and Information Services Center. TCCON data were obtained from the TCCON Data Archive, hosted by the Carbon Dioxide Information Analysis Center (CDIAC) - tccon.onrl.gov. We warmly thank those who made these data available.

**Code/Data availability**

355    The codes used in this paper and the CAMS model simulations are available, upon request, from the author. The OCO-2 and TCCON data can be downloaded from the respective websites.

**Author contributions**

FMB designed the study. LD developed the codes and performed the computations. All authors shared the result analysis.

**Competing interests**

360    The author declare no competing interests.

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

 **Figures and Tables**

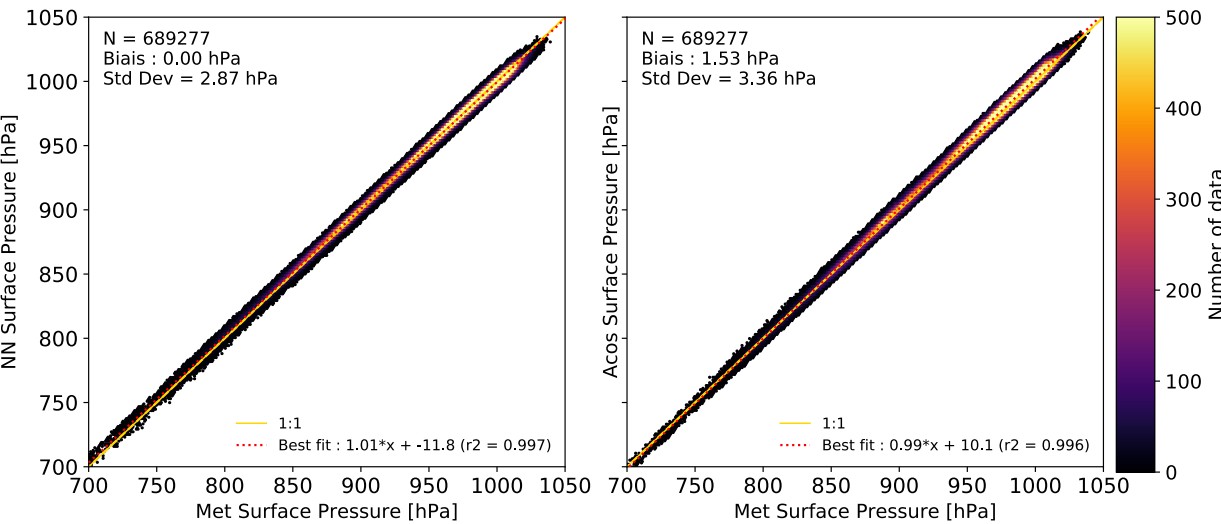

**Figure 1**: Density histogram of the surface pressure retrieved from the OCO-2 satellite measurements against that derived
from GEOS-FP-IT. The left figure is for the NN approach while the right figure is for the ACOS v9r retrieval (using the
official bias-correction). The figure insets provide the number of data points, the bias, the standard deviation, the equation
of the best linear fit and the correlation. The yellow line is the 1:1 line whereas the red dotted line is the best linear fit.

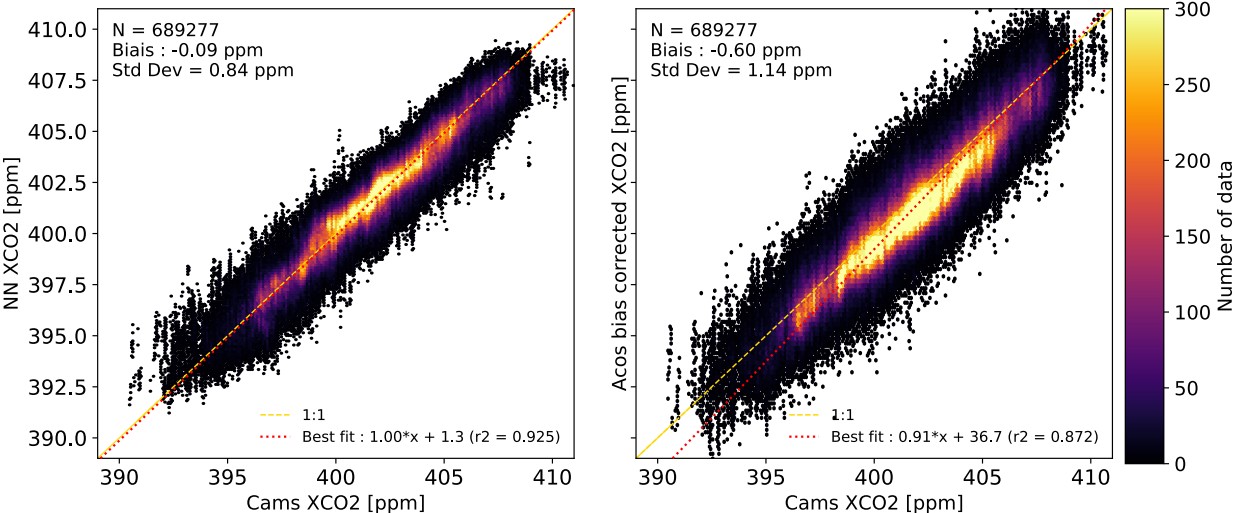


**Figure 2**: Same as Figure 1 but for XCO2. In this case, the reference data is the CAMS v18r2 simulation.

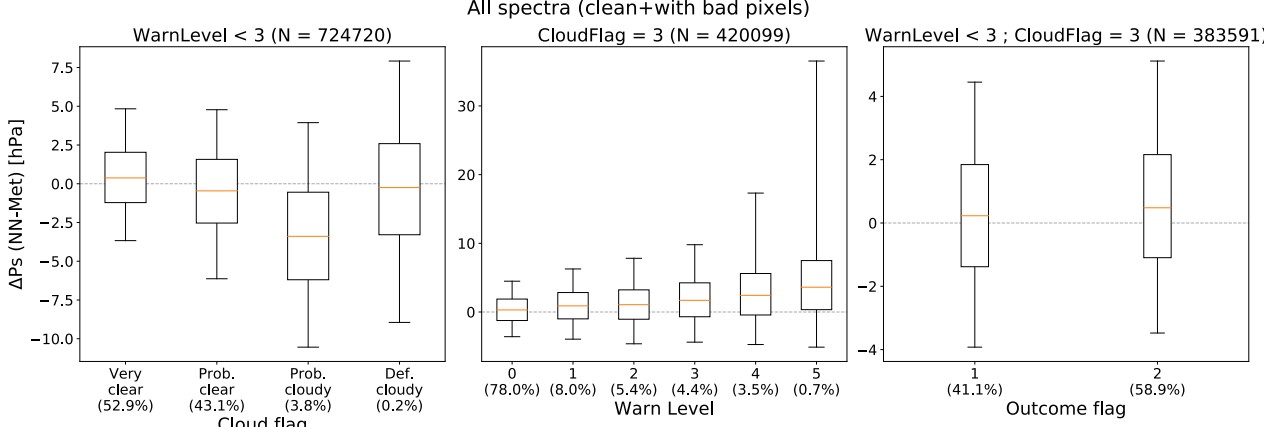

**Figure 3**: Statistics on the difference between the surface pressure retrieved by the NN approach and that derived from the weather analyses, as a function of various quality parameters. In these figures, the red line is the median, the boxes indicate the 25 and 75% percentiles and the whiskers indicate the 5-95% range. The left figure shows the statistics as a function of the cloud flag, the middle figure is as a function of the warn level, while the right figure is as a function of the outcome flag.

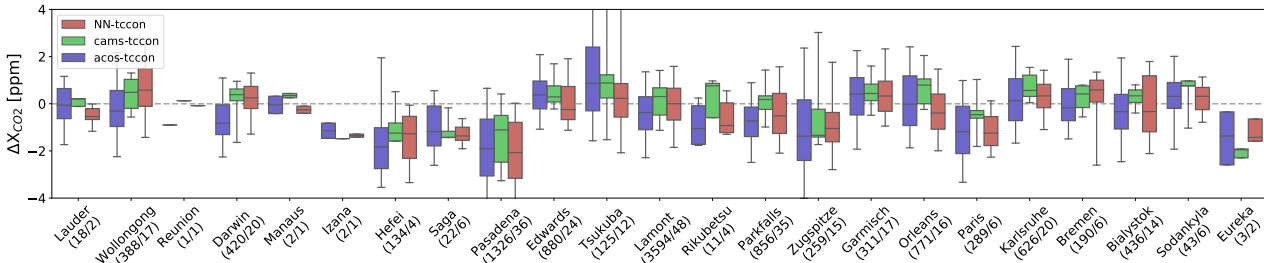

**Figure 4**: Statistics of the differences between the NN retrieval (red), the CAMS model (green) or the bias-corrected ACOS retrievals (blue) and the TCCON retrievals. The boxes indicate the 25-75% percentiles and the median is shown by the horizontal line within the box. The whiskers indicate the 5-95% percentiles. Stations are ordered by increasing latitudes. The numbers below the station name indicate the number of individual observations and coincidence days used for the statistics. The references of the various TCCON observations are provided in table 1.

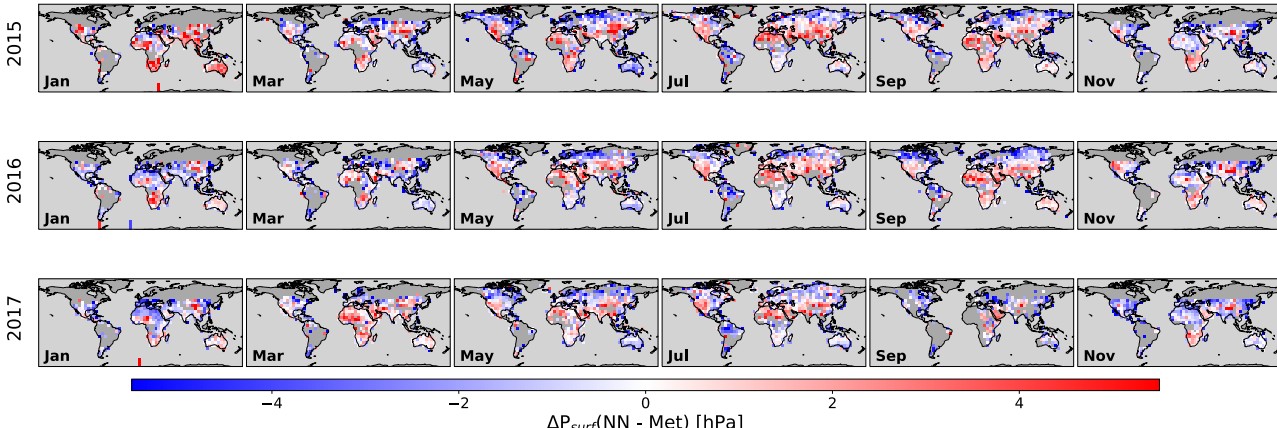


**Figure 5**: Difference between the NN estimate of the surface pressure and the numerical weather analyses. The differences have been averaged at monthly and spatial 5°×5° resolutions. The results are shown for 3 years and only for the months that were not used for the training.


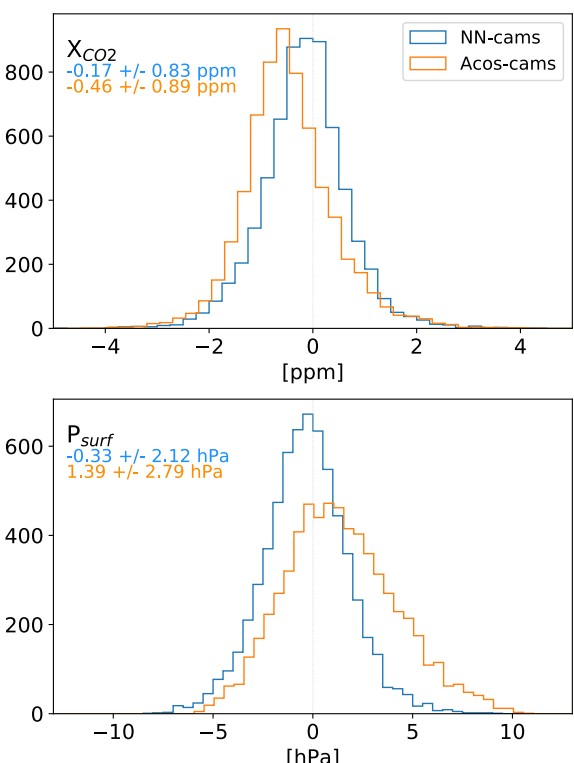

**Figure 6**: Histogram of the monthly mean differences, at 5° resolution (such as those shown in Figure 5), between the satellite retrievals and the CAMS model. The top figure is for XCO2 while the bottom figure is for the surface pressure. The blue line is for the NN product while the orange line is for ACOS.


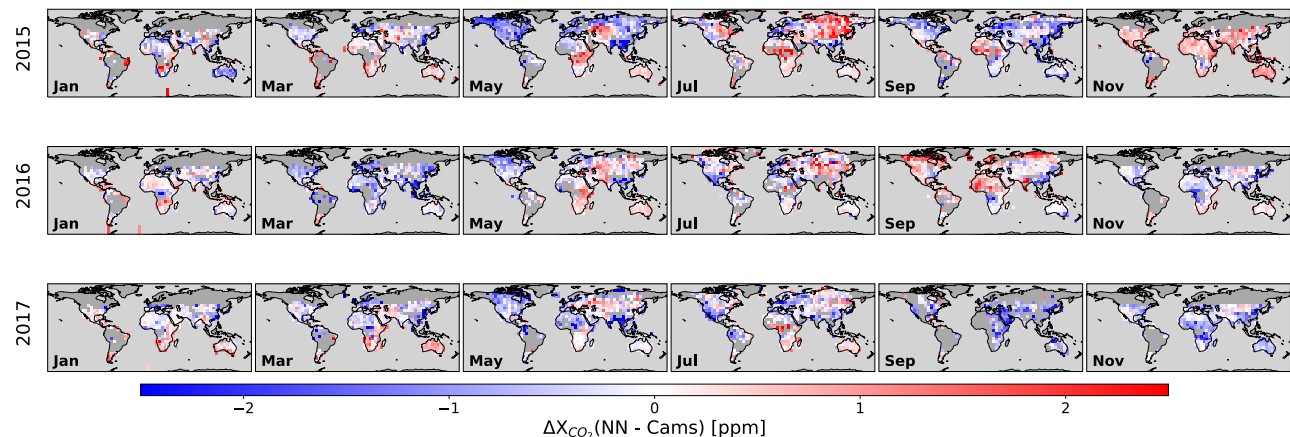

**Figure 7**: Same as Figure 5 but for the difference between the XCO2 estimated by the NN approach and that derived from the CAMS model.

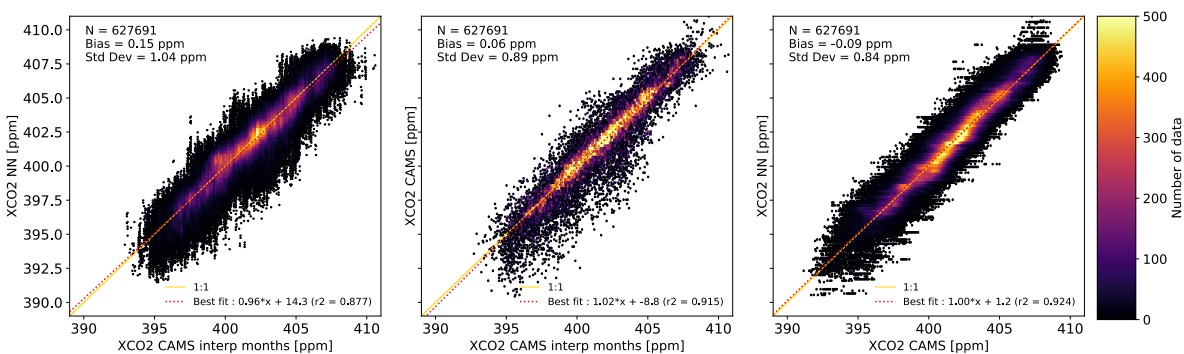

**Figure 8**: Scatter plots of XCO2 estimated by the NN, the CAMS model, and the CAMS model that has been interpolated in time from adjacent months (see text for details). Note that the number of points is less than in Figure 2 because the edge months could not always be interpolated.

**Table 1**: TCCON stations used in this paper (Figure 4). The data have been obtained from the tccondata.org web site at during the summer of 2019.

| Stations | [ lat ; lon ] | Altitude [m] | Reference | Biases NN/ACOS/CAM | Std Dev NN/ACOS/CAM |
|---|---|---|---|---|---|
| Lauder | [ -45.04 ; 169.68 ] | 370 | Sherlock et al. 2017 | -0.48/-0.25/0.076 | 0.43/1.51/0.16 |
| Wollongong | [-34.41 ; 150.88] | 30 | Griffith et al. 2017 | 0.60/-0.20/0.42 | 1.21/1.32/0.60 |
| Reunion | [ -20.90 ; 55.49 ] | 90 | De Maziere et al. 2017 | -0.08/-0.90/0.13 | -/-/- |
| Darwin | [ -12.43 ; 130.89 ] | 30 | Griffith et al. 2017 | 0.19/-0.69/0.23 | 0.80/1.09/0.72 |
| Manaus | [ -3.21 ; -60.6] | 50 | Dubey et al. 2017 | -0.25/-0.05/0.34 | 0.43/1.04/0.26 |
| Izana | [ 28.3 ; -16.48 ] | 2300 | Blumenstock et al. 2017 | -1.35/-1.14/-1.48 | 0.18/0.92/0.0 |
| Hefei | [ 31.90 ; 118.67 ] | 30 | Liu et al. 2018 | -1.47/-1.58/-1.01 | 1.11/1.76/0.63 |
| Saga | [ 33.24 ; 130.29 ] | 10 | Shiomi et al. 2017 | -1.36/-1.03-1.15 | 0.57/1.22/0.59 |
| Pasadena | [ 34.14 ; -118.13 ] | 240 | Wennberg et al. 2017 | -2.12/-1.87/-1.41 | 1.57/1.64/1.17 |
| Edwards | [ 34.96 ; -117.88 ] | 700 | Iraci et al. 2017 | 0.07/0.41/0.50 | 1.00/1.01/0.64 |
| Tsukuba | [ 36.05 ; 140.12 ] | 30 | Morino et al. 2017 | 0.42/1.43/1.05 | 2.13/2.53/1.61 |
| Lamont | [ 36.6 ; -97.49 ] | 320 | Wennberg et al. 2017 | -0.03/-0.38/0.16 | 1.07/1.21/0.94 |
| Rikubetsu | [43.46 ; 1473.77 ] | 390 | Morino et al. 2017 | -0.57/-0.84/0.47 | 0.84/1.07/0.98 |
| Parkfalls | [ 45.94 ; -90.27 ] | 440 | Wennberg et al. 2017 | -0.41/-0.75/0.11 | 1.15/1.01/0.72 |
| Zugspite | [ 47.42 ; 11.06 ] | 2960 | Sussmann and Rettinger 2017 | -0.85/-1.14/-0.83 | 1.45/1.85/1.36 |
| Garmisch | [ 47.48 ; 11.06 ] | 740 | Sussmann and Rettinger 2017 | 0.40/0.28/0.43 | 0.98/1.29/0.62 |
| Orleans | [ 47.97 ; 2.11 ] | 130 | Warneke et al. 2017 | -0.35/0.13/0.66 | 1.06/1.38/0.67 |
| Paris | [ 48.85 ; 2.36 ] | 60 | Te et al. 2017 | -1.29/-1.24/-0.62 | 1.30/1.66/1.23 |
| Karlsruhe | [ 49.1 ; 8.44 ] | 110 | Hase et al. 2017 | 0.26/0.21/0.75 | 0.80/1.29/0.55 |
| Bremen | [ 53.10 ; 8.85 ] | 7 | Notholt et al. 2017 | 0.30/-0.07/0.36 | 1.11/1.02/0.45 |
| Bialystok | [ 53.23 ; 23.02 ] | 180 | Deutscher et al. 2017 | -0.11/-0.32/0.33 | 1.31/1.30/0.42 |
| Sodankyla | [ 67.37 ; 26.63 ] | 190 | Kivi et al. 2017 | 0.26/0.24/0.61 | 0.79/1.36/0.80 |
| Eureka | [80.05; -86.42] | 600 | Strong et al. 2017 | -1.02/-1.50/-2.16 | 1.01/2.25/0.41 |

**Appendix**

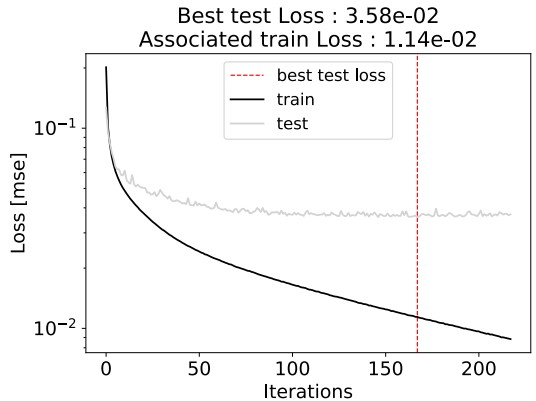

Figure A1: Illustration of the iterative convergence of the NN during its training. The loss is an indicator of the difference between the NN estimate and the dataset. One dataset is used for the best estimate of the NN weights whereas another independent one is used for the evaluation of the NN capability. The NN is stopped when there is no further reduction of the loss for the test dataset for 50 iterations. The weight for the NN are those obtained for the lowest loss of the test dataset (iteration 167 on the figure).

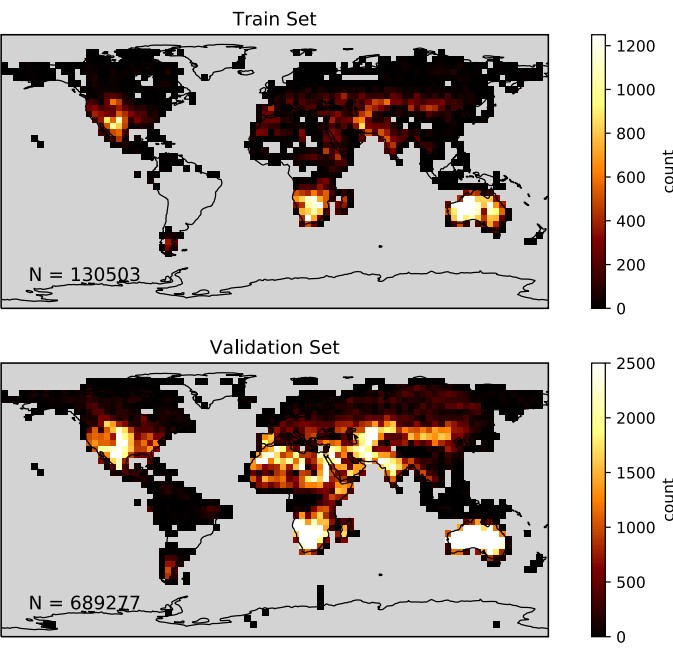

Figure A2: Spatial density of the observation that have been used for the training (top) and validation (bottom) processes.

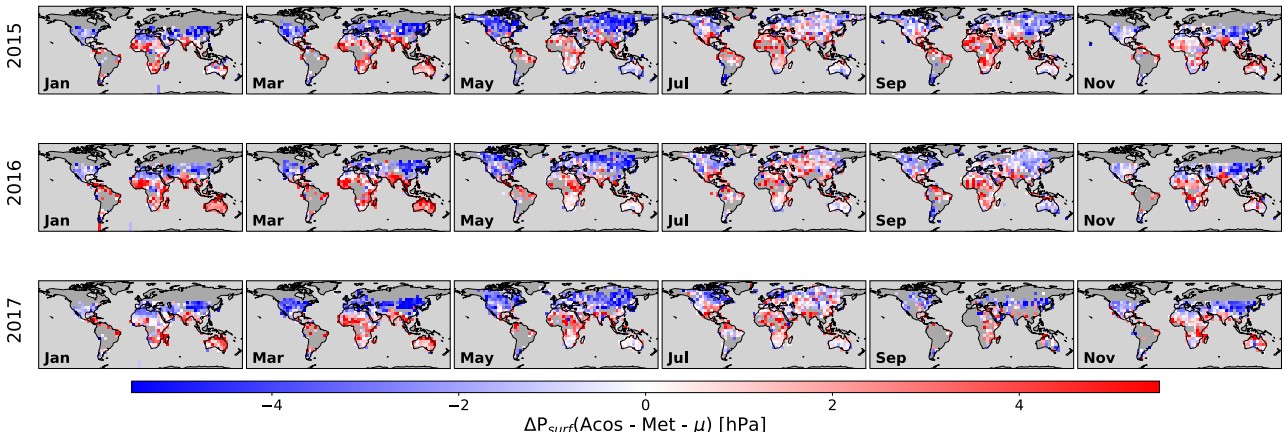

Figure A3: Same as Figure 5 but for the surface pressure retrieved by the ACOS algorithm. The mean bias over the full period (μ) is removed so that the differences are centered on zero.

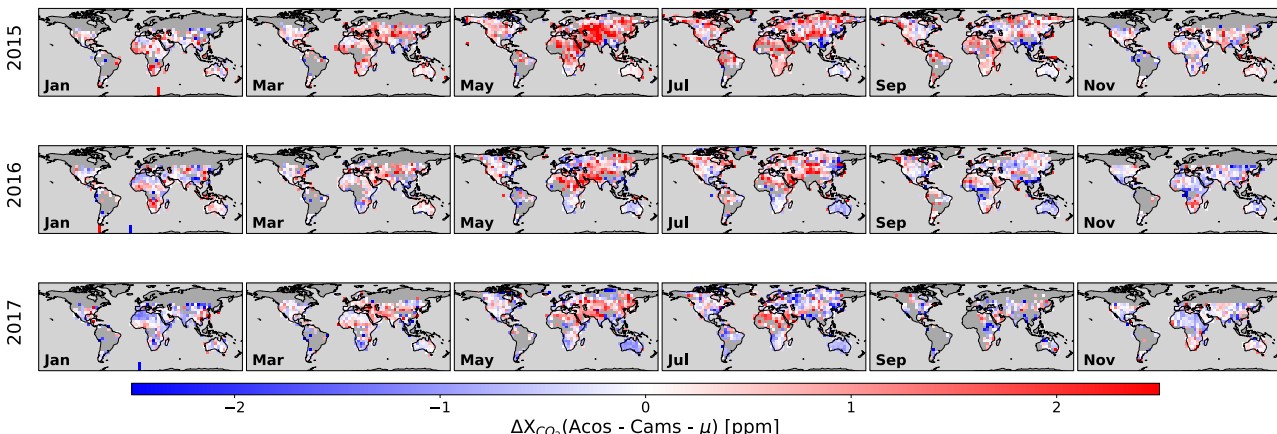

Figure A4: Same as Figure 7 but for the XCO2 retrieved by the ACOS algorithm. The mean bias over the full period (μ) is removed so that the differences are centered on zero.