# Peer review of "XCO2 estimates from the OCO-2 measurements using a neural network approach"

_Atmospheric Measurement Techniques, 2020_

## Referee Comment (RC1) · Anonymous Referee #1 · 7 Jul 2020

The authors describe in their manuscript the usage of an artificial neural network (ANN) to retrieve XCO2 and surface pressure from OCO-2 radiances. The topic fits well to the aims and scopes of AMT and is highly relevant because of the immense computational resources needed to process current and moreover future satellite data with state of the art full physics algorithms which are still prone to biases that require empirical corrections and have unknown origin. However, concluding my general and specific comments below, the presented material does not provide enough evidence to support the main conclusions namely that the results indicate that the ANN approach outperforms the operational NASA full physics algorithm and that it can be used to improve our knowledge of CO2 fluxes. It is understandable to limit the ANN development to simple cases at first (e.g., nadir only) and it is also not to be expected that the ANN

will produce perfect results from the first try on. However, the functionality must be provable. This means it must become clear that the ANN has indeed learned and generalized primarily from the spectral information so that it is able to follow also un-expected CO2 features such as plumes. The presented material seems not sufficient to prove or disprove this. I'm sorry that I cannot give a more positive feedback, but because of this and the many open questions I cannot recommend a publication at AMT. However, due to the relevance of the topic, I encourage the authors to continue their work and, if the results allow it, to resubmit a revised manuscript.

**General comments:**

I have some concerns about the suitability of the input parameters to the ANN. As described in my specific comment L86-L88, potentially important information is rejected from the spectra.

The training data set spans over the same four-year (2015-2018) time period as the test data set and the authors emphasize that this allows XCO2 variations of about 2%. If it is important, that the training covers all possible CO2 concentrations, it is questionable, if a representative training data set can be found suitable also to analyze future OCO-2 data because CO2 is continuously increasing from year to year. What happens when applying the ANN to data from April 2020 including unprecedented large CO2 concentrations due to the continuous year-to-year increase of atmospheric CO2? Would the ANN still give reasonable results when training only with data from 2015?

ANNs are more or less black boxes in the sense, that it is not easily possible to find out which physical relationships they have learned. As CO2 is well mixed and longlived, it is easy to make a relative good guess of its concentration without using any measurements. As an example, by estimating XCO2 only from latitude and time one can achieve already a good agreement with TCCON. Nevertheless, nothing new can be learnt about CO2 from such estimates. Therefore, it is crucial to prove that the ANN's XCO2 is primarily coming from the absorption depth within the spectra. This,
however, has not been done. One could confront the ANN with simulated spectra and show that it is indeed able to follow the simulated variations of the CO2. Such simulations could also be used to derive an estimate for the column averaging kernels, e.g., for different solar zenith angles. Additionally, one could apply the ANN to small scenes (with basically constant observation geometry) from which it is known that they include isolated CO2 plumes (see Nassar et al. (2017), or Reuter et al. (2019) for examples). Than it could be analyzed in how far the ANN is capable to follow the XCO2 enhancements and in how far the results agree with those of ACOS.

There are indications for over fitting: as the authors state, the maps in Fig.5 show biases in the test data set which do not exist for the training data set.

Conventional full physics algorithms allow (and usually require) post filtering by analyzing, e.g., the spectral fit quality but also quantities such as the posteriori XCO2 error estimate. Is this also possible for an ANN approach?

The TCCON validation seems to be not state of the art. For example, it does not consider the averaging kernels. For the ANN this is because they have not been computed, but for the TCCON and ACOS they are available. Which metrics have been used to quantify the performance? Usually, the average single sounding precision and the station-to-station biases are computed as a minimum set of parameters describing the quality. Please note that a low scatter could potentially also be observed when simply ignoring the satellite data which always add instrumental noise. This should also be considered/discussed when comparing CAMS (including no instrumental noise) with TCCON.

Please specify what is meant with the different measures of agreement that are used throughout the manuscript (trueness, accuracy, precision, skill, quality).

Specific comments:

L19: Please define what exactly is meant with precision (at least within the main text).

**AMTD**
Is it, e.g., the standard deviation of the retrieved values or the standard deviation of the difference to a truth?

L42-L44: "Similarly..." This paragraph reads like a description of the light path proxy method, e.g., used by Schneising et al. (2008). However, the idea of full physics algorithms is fundamentally different to this. Because of wavelength dependencies of the surface reflection and of the scattering properties (e.g., optical thickness), the light path is different in the O2 and CO2 bands. Therefore, full physics algorithms use the O2 band to infer knowledge on the scattering particles or processes which allow to estimate the light path in the CO2 band from measurements in the O2 and CO2 band (see, e.g., Butz et al. (2011), Cogan et al. (2012), O'Dell et al. (2018), Reuter et al. (2017), Yoshida et al. (2013)). However, it is correct, that some (not all) full physics algorithms also retrieve the surface pressure.

L45: Molecular (Rayleigh) scattering is not a main difficulty as it is well known. The main problems are aerosols and clouds (see publications cited in my last comment).

L49: Please describe what is meant with "have been optimized".

L53: Why do you consider the signal to be the deviation from the prior? Shouldn't the signal be rather the actual variability.

L65-L67: RT models can simulate the radiance usually extremely accurate. However, the input to the RT models (e.g., unknown scattering phase functions, surface BRDF) and approximations needed to meet the requirements on the computational efficiency are the problems. Additionally, there may be unknown instrumental effects (uncertainties in the instrumental line shape function, polarization sensitivity, stray light, etc.).

L71: "The evaluation results show ...." should be moved to the discussion.

L76: Why footprint #4 not #7 or #3? Do the results critically depend on the used footprint?

L77: Please discuss if this issue principally will render ANN approaches impossible for
glint observations and if not, outline potential solutions.

L80: Have you attempted to remove/mask the most affected pixels?

Sec.2: Please describe which OCO-2 data exactly has been used. Which version and where can be obtained from?

L86-L88: Dividing by the maximum is potentially not ideal because this maximizes the influence of instrumental noise or outliers due to cosmic rays and, additionally, it does not account for slopes in the spectrum. Such effects could be reduced by, e.g., dividing by the 90% percentile of the, e.g., 100 left-most spectral points. However, my main concern here is another: Dividing by the maximum radiance removes important information from the spectrum. Namely the information on albedo (as mentioned in the manuscript). As discussed in the literature (see provided references of full physics algorithms), unknown scattering properties introduce among the largest uncertainties in XCO2 retrievals. Knowledge of the albedo is important to infer knowledge of scattering properties. Consider an atmosphere with a surface pressure of 1000hPa and a scattering layer at 500hPa, reflecting 1% or the incoming radiation. Let CO2 absorb 80% of the radiance along the light path (sun-surface-satellite). This means about 40% will be absorbed along the light path sun-scattering layer-satellite. In the case of an albedo of 100%, the average absorption would be only slightly less than 80%. In the case of an albedo of 0%, the average absorption would amount 40%. This means, the relative depth is not a good measure for the number of particles in the total column. If you would normalize by the solar incoming radiance instead of the maximum radiance you would retain information on albedo, and therefore, also on scattering. Additionally, it shall be noted that, the light path in the O2 band can significantly differ from that in the CO2 band because of differences in the albedo and scattering properties.

L91: The influence of the azimuth should be discussed in the results section.

L95: 2557 input neurons are quite a lot and results in a rather complex ANN. Often one tries to reduce the dimensionality of the input data by performing, e.g., a PCA. This can
probably also help the ANN to generalize instead of memorize. Why have you decided to use the full dimensionality of the input?

L96: Why 500 hidden neurons? Is there a rule of thumbs to select a suitable number of hidden neurons? Please discuss how your results depend on the complexity of the network topology. Do you use a so called bias neuron?

L100: Preventing the ANN from over training is certainly important. However, I'm not sure if it is a good idea to stop iterating before convergence is reached. Could the fact that overtraining happens for more than 100 iterations hint at a too complex network topology. Would it be an option to prevent from over fitting by choosing a less complex network with fewer hidden neurons? Additionally, a plot showing the convergence behavior would be nice to have (e.g., RMSD performance of the training and test data vs. number of iterations).

L103-L104: It is an important point whether the ANN uses information of time and position of the observation or not. Therefore, please discuss, in how far the observation geometry can provide the ANN indirectly with information on the position and/or time of the observations. Which parameters do you mean with "observation geometry" (is, e.g., surface elevation included)?

L104-L106: CO2 does not only have a seasonal variation, but it is also continuously increasing from year to year. Therefore, when having in mind a potential application to future data, the ANN will usually have to deal with concentrations larger than used for the training. How would the ANN behave in such cases? This could, e.g., be answered by confronting the ANN with RT simulated radiances.

L113: Why do you use ppm as unit for surface pressure?

L123: As mentioned in L121, surface based measurements have been used.

L125: Model pressure usually includes water vapor. However, XCO2 is the dry-air column-average. Therefore, strictly speaking, you would have to compute the weights
according to the pressure difference corrected for the water vapor content.

L128-L137: Interpolating only within the sorted spectrum is a good idea. However, the surface reflectance can introduce significant slopes within the spectra which may significantly change the rank of the spectral pixels. How large is the impact of this effect?

L166-L171, Fig.3: Why do you use for Fig.3 (left) only soundings where an ACOS retrieval has been made? This drastically reduces the number of cloudy soundings. From Fig.3 (left), I would estimate, that the ANN is capable to derive the surface pressure for definitively cloudy cases nearly bias free with a standard deviation of better than 5hPa. In cases of clouds that are not optically extremely thin, the spectra should not include significant information on the surface pressure. Additionally, the light path is shortened which can usually be interpreted as low surface pressure. Please discuss why the ANN is still capable to derive the surface pressure so well.

L175: I would suggest to also show the corresponding figure for XCO2. If it turns out that the ANN is also capable, to derive XCO2 in definitively cloud contaminated scenes, I would also suggest to add a discussion where this information is coming from.

L181-L190: How have you accounted for the column averaging kernels? At least for ACOS and TCCON, this information is available and should be used. Additionally, the usage of TCCON data should be described in Sec.2 and it should be mentioned in the main text, where the TCCON data can be obtained from and when they were downloaded.

L206-L210: I have some concerns with this arguing. If you chose a very complex network topology, the ANN might be well able to reproduce CAMS (including systematic persistent biases). If the ANN is too simple, it may not be able to follow the actual CO2 variability. I have the impression that this paragraph implicitly assumes that the network complexity is "just right" so that the ANN was not able to learn biases of CAMS but still generalized enough to follow the actual CO2.

AMTD
L214: Please discuss why you observe a more or less persistent bias pattern in the even months but not significant differences in the odd months (used for the training). The fact that the training data set performs significantly better than the test data set usually hints at over fitting.

L215: In L206-L210 you suggest that XCO2 difference may come from model deficiencies. Why do you interpret the surface pressure differences as ANN biases?

Fig.5, 7, A1, A2: Please also show the even month used for the training because significant differences between even and odd months, can hint at a potential over fitting. The shown differences are in the order of 0.5%. What is the expected impact of neglecting the averaging kernels?

L237: The presented material does not allow this conclusion. Please, particularly, see my specific comments related to the validation method, the used input data for the ANN, and the lack of a prove that the ANN's XCO2 variability is indeed primarily coming from the spectral information.

L238-L239: Please define precision. For which product there is no independent truth (surface pressure or XCO2)?

L260-L263: I would have concerns with both options: i) If the OCO-2 spectra do not include information on, e.g., the upper most CO2, the ANN's AKs will have no sensitivity here independently from the used training data. ii) AKs usually differ from L2 algorithm to algorithm. I would suggest to compute typical AKs by confronting the ANN with simulated spectra for different observation geometries.

Technical corrections:

L11,43: column integrated CO2 dry air mole fraction -> column-average dry-air mole fraction of CO2

L13: uses a full -> is a so called full
L38: During -> Along

L63: Comprehensive -> Representative

L124: observations -> observation

Fig1: Please add a legend for the dashed/dotted yellow and red lines. The visibility of these lines is poor. The caption or the axis should include the information which models have been used?  $r^2 -> r^2$

- L182, L184: TCCON network -> TCCON
- L183: Fourier Transform Infrared -> Fourier transform infrared
- L183: tuned against -> calibrated with
- L184: "tuning" do you mean "bias correction"?
- L186: "neither...or" do you mean "either ... or"?
- Fig.3: Please increase the font size.

Fig.4: Please increase the font size. TCCON station names should start with upper case letters.

Fig.5, 7, A1, A2: The font size is too small, white is ambiguous (snow/ice or delta\_p = 0), green is not explained.

References:

Butz et al. (2011): Butz, A., Guerlet, S., Hasekamp, O., Schepers, D., Galli, A., Aben, I., Frankenberg, C., Hartmann, J.-M., Tran, H., Kuze, A., Keppel-Aleks, G., Toon, G., Wunch, D., Wennberg, P., Deutscher, N., Griffith, D., Macatangay, R., Messerschmidt, J., Notholt, J., and Warneke, T.: Toward accurate CO2and CH4observations from GOSAT, Geophys. Res. Lett., 38, L14812, https://doi.org/10.1029/2011GL047888, 2011
Cogan et al. (2012): Cogan, A. J., Boesch, H., Parker, R. J., Feng, L., Palmer, P. I.,Blavier, J.-F. L., Deutscher, N. M., Macatangay, R., Notholt, J.,Roehl, C., Warneke, T., and Wunsch, D.: Atmospheric carbondioxide retrieved from the Greenhouse gases Observing SATel-lite (GOSAT): Comparison with ground-based TCCON obser-vations and GEOS-Chem model calculations, J. Geophys. Res.,117, D21301, https://doi.org/10.1029/2012JD018087, 2012

O'Dell et al. (2018): O'Dell, C. W., Eldering, A., Wennberg, P. O., Crisp, D., Gunson, M. R., Fisher, B., Frankenberg, C., Kiel, M., Lindqvist, H., Man-drake, L., Merrelli, A., Natraj, V., Nelson, R. R., Osterman, G. B., Payne, V. H., Taylor, T. E., Wunch, D., Drouin, B. J., Oyafuso, F., Chang, A., McDuffie, J., Smyth, M., Baker, D. F., Basu, S., Chevallier, F., Crowell, S. M. R., Feng, L., Palmer, P. I., Dubey, M., García, O. E., Griffith, D. W. T., Hase, F., Iraci, L. T., Kivi, R., Morino, I., Notholt, J., Ohyama, H., Petri, C., Roehl, C. M., Sha, M. K., Strong, K., Sussmann, R., Te, Y., Uchino, O., and Ve-lazco, V. A.: Improved retrievals of carbon dioxide from OrbitingCarbon Observatory-2 with the version 8 ACOS algorithm, At-mos. Meas. Tech., 11, 6539–6576, https://doi.org/10.5194/amt-11-6539-2018, 2018

Nassar et al. (2017): Nassar, R., Hill, T. G., McLinden, C. A., Wunch, D., Jones, D., and Crisp, D.: Quantifying CO2 emissions from individual power plants from space, Geophys. Res. Lett., 44, 10045–10053, https://doi.org/10.1002/2017GL074702, 2017

Reuter et al. (2017): Reuter, M., Buchwitz, M., Schneising, O., Noël, S., Rozanov, V., Bovensmann, H., and Burrows, J. P.: A Fast AtmosphericTrace Gas Retrieval for Hyperspectral Instruments Approximat-ing Multiple Scattering – Part 1: Radiative Transfer and a Po-tential OCO-2 XCO2Retrieval Setup, Remote Sens., 9, 1159,https://doi.org/10.3390/rs9111159, 2017

Reuter et al. (2019): M. Reuter, M. Buchwitz, O. Schneising, S. Krautwurst, C.W. O'Dell, A. Richter, H. Bovensmann, and J.P. Burrows: Computation and analysis of atmospheric carbon dioxide annual mean growth rates from satellite observations during
2003-2016, Atmos. Chem. Phys., https://www.atmos-chem-phys.net/19/9371/2019, 2019

Schneising et al. (2008): O. Schneising, M. Buchwitz, J. P. Burrows, H. Bovensmann, M. Reuter, J. Notholt, R. Macatangay, T. Warneke: Three years of greenhouse gas column-averaged dry air mole fractions retrieved from satellite - Part 1: Carbon dioxide. Atmospheric Chemistry and Physics, doi:10.5194/acp-8-3827-2008, 8, 3827-3853, 2008

Yoshida et al. (2013): Yoshida, Y., Kikuchi, N., Morino, I., Uchino, O., Oshchepkov, S.,Bril, A., Saeki, T., Schutgens, N., Toon, G. C., Wunch, D., Roehl,C. M., Wennberg, P. O., Griffith, D. W. T., Deutscher, N. M., Warneke, T., Notholt, J., Robinson, J., Sherlock, V., Connor, B., Rettinger, M., Sussmann, R., Ahonen, P., Heikkinen, P., Kyrö, E., Mendonca, J., Strong, K., Hase, F., Dohe, S., and Yokota, T.: Improvement of the retrieval algorithm for GOSAT SWIRXCO2and XCH4and their validation using TCCON data, At-mos. Meas. Tech., 6, 1533–1547, https://doi.org/10.5194/amt-6-1533-2013, 2013

**AMTD**

---

## Short Comment (SC1) · 12 Jul 2020

We thank the reviewer for her/his in-depth analysis of our paper. The present document is not a full answer to his/her review (which we will make when the other review becomes available), but only an answer to the main comment that we have not demonstrated that the ANN primarily uses the spectral information to estimate the XCO2:

"This means it must become clear that the ANN has indeed learned and generalized primarily from the spectral information so that it is able to follow also un-expected CO2 features such as plumes."

As stated in the manuscript, the only variables that are provided to the ANN are the measured spectra and the observation geometry. The observation location, year and

date within the year are NOT input to the ANN, in contrast to what the reviewer's text may suggest (p. C2, l. 4 from bottom). The observation geometry is the solar and view zenith angles, together with the relative azimuth. It does not include the target surface elevation (reviewer's p. C6). The OCO-2 instrument follows a complex attitude pattern with respect to the solar principal plane but it is the same over the longitudes and from one year to the next. Thus, the ANN cannot find any information on the observation longitude and year of observation from the observation geometry. It is then clear that the ANN retrieves XCO2 primarily from the spectral information, as there is nothing else available as input. The XCO2 variations along the longitude, and even more from year to year, are actually much larger than the ANN error that we estimate show on Figure 2. Figure 2 demonstrates that the ANN is able to reproduce variations of XCO2, as it does for the variations of the surface pressure (Figure 1).

We believe this is a clear demonstration that the ANN retrieves XCO2 from the spectral information.

The reviewer requires the demonstration that the NN can detect unexpected XCO2 features such as plumes (p. C3, top). Let us recall that there is no available "truth" for such plume so that it can hardly be a demonstration of the data validity. Indeed, there is evidence of "false" plumes in the ACOS XCO2 dataset that appear to be generated by aerosol or surface albedo features. It is certainly a valid objective to analyse the small-scale variability of the ANN retrievals but it requires the analysis of all footprints rather than a single one (FOV #4) as done in the present paper, and is therefore a forthcoming step.

Also, the reviewer suggest that we should apply the ANN to simulated spectra (p. C3, top). We fully disagree with this suggestion. Although the radiative transfer models are very accurate, as the reviewer writes, they do show significant systematic differences with the observation. The ACOS algorithm uses empirical EOF to account for these differences (in addition to bias correction). The ANN training accounts for these systematic differences (i.e. it is fitted to the measurements, not to some simulated spectra). Applying this ANN to radiative transfer simulations that generate biased spectra can only result in poor results.

---

## Referee Comment (RC2) · Anonymous Referee #1 · 14 Jul 2020

*"It is then clear that the ANN retrieves XCO2 primarily from the spectral information, as there is nothing else available as input."*

I just trained a multi-layer perceptron ANN with a hidden layer of 50 neurons. Input: Solar zenith angle, viewing zenith angle, and azimuth difference of footprint#4 nadir OCO-2 soundings of the year 2015. Output: Latitude, CarbonTracker XCO2. The predicted latitude has a precision of about 6° (correlation 0.98) and the predicted XCO2 has a precision of 1.8ppm (correlation 0.67).

This means that the used observation geometry implicitly includes a lot information on the position (note that one of the largest signals in XCO2 is the latitudinal gradient) and, therefore, also on the typical XCO2 distribution. Without using any spectral information, it is possible to explain about 45% of CarbonTracker's XCO2 variance in 2015. Any

additional information from the spectral bands (even if no CO2 bands are included) has the potential to add context improving the average XCO2 prediction performance.

I'm not saying that the ANN presented in the paper indeed did not learn to retrieve CO2 primarily from the CO2 line depth but the presented material is also not sufficient to prove that it did.

*"Let us recall that there is no available truth for such plume so that it can hardly be a demonstration of the data validity. Indeed, there is evidence of false plumes . . ."*

Please provide a reference if there is a peer-reviewed publication discussing these false plumes. In my review, I already cited the publications of Reuter et al. (2019) and Nassar et al. (2017) showing some plumes. Are there good reasons to consider all these false plumes? They are broadly consistent with model wind fields, S5P NO2, and emission databases. They include observations in nadir mode and are rather data-dense so that the signal should be visible in footprint#4.

*"Also, the reviewer suggest that we should apply the ANN to simulated spectra. We fully disagree . . ."*

It would be sufficient to select a more or less arbitrary OCO-2 spectrum and simulate only the XCO2 Jacobian and add it to the spectrum. The differences between simulated and measured spectra have a large systematic component (that's why they can be fitted with few EOFs). Therefore, the simulated Jacobian should agree very well with the actual Jacobian. Additionally, it shall be noted, that there are non-NASA full physics algorithms for GOSAT and OCO-2 which produce reasonable results without fitting EOFs. I.e., they deal with the differences between measured and simulated radiances without messing everything up.

**References**

Nassar et al. (2017): Nassar, R., Hill, T. G., McLinden, C. A., Wunch, D., Jones, D., and Crisp, D.: Quantifying CO2 emissions from individual power plants from space, Geophys. Res. Lett., 44, 10045–10053, https://doi.org/10.1002/2017GL074702, 2017

Reuter et al. (2019): M. Reuter, M. Buchwitz, O. Schneising, S. Krautwurst, C.W. O'Dell, A. Richter, H. Bovensmann, and J.P. Burrows: Towards monitoring localized CO2 emissions from space: co-located regional CO2 and NO2 enhancements observed by the OCO-2 and S5P satellites, Atmos. Chem. Phys., https://www.atmos-chem-phys.net/19/9371/2019, 2019

---

## Short Comment (SC2) · 20 Jul 2020

Again, we realy do appreciate the reviewer involvement in the evaluation of our work.

Unfortunately, we have not been able to reproduce the reviewer results concerning the XCO2 correlation (between true and ANN retrieval) when using only observation geometry data. We do find, however, a similar RMS (1.8 ppm). This RMS is for a single year of observation. When using 3 years of observations, as done in the paper, it is even larger (2.64) which is not surprising as the XCO2 growth rate impacts the data variability while there is absolutely no information in the input data to infer the year of observation (and thus the year to year variability of XCO2).

Conversely, when the spectra are provided as input to the ANN, we get a RMS of 0.7

ppm (for the 3 year period).

Attached is a figure that shows the latitude-temporal evolution of the SZA, Asimuth and XCO2. Clearly, the SZA and Azimuth patters are reproduced from year to year, while XCO2 shows a significant change. Also, not that there is no longitude information in SZA and Azimuth (ie their values vary as a function of latitude and time in the year, but nothing else, contrarily to XCO2)

This is a clear demonstration, we feel, that the bulk of information is in the spectra rather than in the observation geometry
* * *
[Figure]

[Figure]

**Fig. 1.**

---

## Referee Comment (RC3) · Christopher O'Dell (Referee) · 27 Jul 2020

Review of David et al., "XCO2 estimates from the OCO-2 measurements using a neural network approach.", by Chris O'Dell.

This work details a fast, artificial neural network (NN) approach to retrieving surface pressure and the column-mean dry air mole fraction of CO2 (XCO2) from high-spectral resolution measurements in the near-infrared from the Orbiting Carbon Observatory-2 (OCO-2). Tradiationally, the most accurate XCO2 retrievals have been from semi-physical ("Fullphysics") retrievals. These are typically iterative, and typically include accurate calculations of multiple-scattering from thin layers of clouds and aerosols, which makes them exceedingly slow. They also tend to be subject to importance biases (of order 1 ppm) due to forward model errors (such as spectroscopic or instrument calibration). A neural-network approach is extremely appealing because it automatically solves the speed problem (NN's are very fast, likely tens of milliseconds per retrieval, vs. minutes for a typical FP approach), and may solve some of the bias problem as well, because they simply train on the "right answer", and do not have to know the details of spectroscopy or instrument models.

**General Comments:**

This work is the first serious effort to use NN's as applied to the XCO2 retrieval problem, and the author's mostly do a good job. However, there are a number of weaknesses and methodological problems in this work that need to be strengthened before I can recommend publication. I see this is a "major revision", but ultimately I believe this work can and should be published, as I believe (for the reasons above) that NN's hold great promise as applied to the XCO2-from-satellites problem.

Most significantly, as the first reviewer pointed out, it is difficult to ascertain from the manuscript alone exactly what the NN algorithm has learned. The authors trained it on a model (CAMS), and as a first validation, tested it against the same model. Using alternating months as training vs. testing is helpful, but the model certainly has deficiencies that are persistent longer than a month in certain regions, so testing well vs. that same model simply is not a validation. The only other real validation given is against TCCON, which seemed to perform well but because of TCCON's lack of good global coverage, again it is very hard to tell how well the model performs globally in any real sense. I am also worried about having to re-train the model every year to deal with the ~2 ppm secular increase, which over a mere 4 years is roughly equal to the entire global variability of XCO2. I believe it would make their argument much stronger if they ran their algorithm on a few select powerplant cases where the enhancement in the plume is reasonable well-constrained. This is possible for power plants with good bottom-up emissions esimates, and cases where the wind speed is reasonably well-known. See for instance Nassar et al. (2017) for some sample cases. If the NN doesn't see a plume at all, we know that it hasn't been properly trained; if it does, it will dramatically bolster the arguments in this work. Arguing that just because the NN doesn't have direct access to location or time information does not mean it cannot indirectly learn other relationships that allow it to appear to learn well. This is a hypothesis, not a proof that it has learned what you think it did.

Related to this, I'm somewhat concerned by training on CAMS and then considering to use the resulting XCO2 to correct CAMS. To show that your method works, you almost need to run a full (and fairly complicated) OSSE where you have a true world, a CAMS-like model world with some CO2 errors (spatially and temporally correlated), train on the latter, and then see if you can recover the former with the NN. I don't see how this is guaranteed to work, honestly. How do you know that you won't somehow reproduce systematic errors in CAMS by using the NN approach? You state in the text that you tacitly assume that CAMS errors are not correlated OCO-2 spectra in given areas and for given months. But because the CAMS errors (likely of order 1 ppm) are of a similar magnitude as the XCO2 signal, it is important to point out that this is merely an assumption, and more extensive validation (or a detailed OSSE study) is necessary to prove it.

Also, you claim to use the "ACOS cloud flag", which you say has values 0, 1, 2, or 3, as a way to define both your training and testing data sets. I think you mean "PreprocessingResults/cloud flag idp" in the L2Std file. If this is correct, please know that this flag is little used by the community. In fact, I've never heard of anyone using it, actually. It was defined about a decade ago for GOSAT and not really touched since then (I verified this with the author of the code that defines it). It has never been carefully validated and it appears to be extremely restrictive ("co2 ratio idp" must be between 0.99 and 1.01 to pass, which is extremely restrictive and appears to cut out entire regions of the globe). Further, using outcome flag=1 is also quite restrictive. Can you please comment on these flags, and why you didn't use the far simpler ACOS xco2 quality flag, which is widely used by the community and is the generally adopted quality flag to use? In the plot below, I have attempted to show the differences between the two approaches for May 2016. I had to match the L2Std files (v8r) to the Lite files (v8), so there may be some differences to what you used in your work, but the general conclusion is that you miss a great deal of data with the highly restrictive data set you are working with. Thus, because it is so restrictive, it may be a far easier task than what ACOS tries to do, which is get the best error possible for the xco2 quality flag = 0 dataset, which is roughly 6 times bigger.

Finally, in section 2 please give the sources of ACOS/OCO-2 data you used with more specificity. What specific versions and datasets of OCO-2 did you use? V8r, or just V8? Did you use L2Std files, L2Dia files, Lite files, etc?

What you train on is pretty critical. I think you should at the very least show a sounding density map of your training (and testing) set. Further, I think you should carefully explain your reasoning on how you choose the filtering. You must at least mention the xco2\_quality\_flag, and ideally you would retrain (or at least test) using this, if you aren't going to define your own quality flags. If you choose to train using very restrictive (clear-sky conservative) filters, please explain this is more detail.

Also, both outcome\_flag and warn\_level (which you use for filtering) come from the ACOS L2FP product (cloud\_flag\_idp comes from a fast, preprocessor code, the IMAP-DOAS Preprocessor, or IDP). It would be much better if you could avoid this entirely, because currently you are throwing away all the soundings that didn't converge or were skipped by the ACOS team, which relies on all the peculiarities of our specific algorithm. To make a useful NN algorithm, it ultimately must be independent of any full physics algorithm, unless you want to *train* on soundings that pass some smaller subset of data that includes L2FP quality flags, but *test* on a more complete set of soundings that doesn't use any L2FP quality flags. But you do not appear to do this.

**Specific Comments:**

L20, Abstract: I don't think TCCON is a "sunphotometer". That kind of implies more moderate resolution measurements. How about "reference ground-based spectrometer" or something similar?

L80: Please clarify "a limited set of spectral elements". Make clear these are the solar (Fraunhofer) lines you're talking about.

L85 (near there) : Do you try to mask deep solar lines as well (to remove their Doppler effects)? Please clarify, with a why or why not.

L90 (near there): Might you include the polarization angle directly to the NN, in addition to / instead of the relative azimuth? That might work even better.

L96: As the readers of this article are likely not NN experts, please discuss the pros and cons of # of hidden layers vs. # of neurons. Also please discuss how was the number 500 chosen or optimized.

L113: I think you mean "1 hPa", not "1 ppm".

L125: XCO2 is defined as weighted by the number of dry air molecules per square meter in each layer, not the pressure width. This can be shown to be roughly proportional to dP \* (1-q) for a given layerm (e.g., O'Dell et al., 2012), where dP is the pressure width and q is the specific humidity in kg/kg. Please recalculate your model XCO2 using this more standard formulation, if possible, or defend your non-traditional XCO2 definition. The differences are generally small (tenths of a ppm), so it is defensible, but if you can be correct, it is best to do so.

L138/Figure 1: This is supposedly for the evaluation dataset, but includes N=381k soundings? In section 2, you say the evaluation dataset only includes 155k soundings. So something is wrong – please explain or fix.

L138-152: Based on Fig 5, there appears to be some problem in the surface pressure retrieval over mountains, specifically a high bias generally in these regions (visible over the Tibetan Plateau and the U.S. Rocky Mountains). Please discuss. I suggest that including the surface elevation in the NN may be a good idea, though technically it shouldn't be necessary.

Regarding TCCON comparison: It would be useful to include the following statistics for ACOS, NN, and CAMS vs. TCCON: Overall Mean, Overall Std Dev, and Stddev of Station mean biases. These are useful to evaluate accurate vs. TCCON in simple statistics. See for example Fig 18b in O'Dell et al (2018). It shows a mean bias of Nadir Land observations vs. TCCON of 0.30 ppm, and a stddev of 1.04 ppm (it has not calculated the stddev of the station-level mean biases; some groups do this, others not). Finally, it doesn't look like you're applying the averaging kernel (AK) correction when comparing ACOS to TCCON. This typically makes the stddev about 0.1 ppm better. If you do not make this correction, please point this out in the text.

Fig4: Please include a horizontal dashed line so we can see the zero-level. Also, please be clear in the caption or the text if the ACOS and NN are sounding-matched. Typically, when we

compare ACOS to TCCON, which use all xco2\_quality\_flag=0 data. If you were to do this, it may change your results for ACOS vs. TCCON (though better vs. worse, I'm not sure).

**Technical/Grammatical:**

- L92: "Although, the NN technique"  $\rightarrow$  "Although the NN technique"
- L124: "For each OCO-2 observation"
- L129: "cosmic flux anomaly"  $\rightarrow$  "cosmic ray flux anomaly"
- L151: "lowest pressure"  $\rightarrow$  "lowest pressures"
- L181: please replace "classical" with "standard" or "traditional"
- L250: "that are described in this paper."

---

## Author Comment (AC1) · 28 Aug 2020

**Answer to reviews**

We warmly thanks both reviewers for their work, their careful analysis, and their suggestions that, we hope, led to an improve version of the manuscript.
We agree with most comments, but not all of them as detailed in the following.

In the following, the reviews are in Italic whereas our answers are in plain text

**Reviewer 1**

*The authors describe in their manuscript the usage of an artificial neural network (ANN) to retrieve XCO2 and surface pressure from OCO-2 radiances. The topic fits well to the aims and scopes of AMT and is highly relevant because of the immense computational resources needed to process current and moreover future satellite data with state of the art full physics algorithms which are still prone to biases that require empirical corrections and have unknown origin. However, concluding my general and specific comments below, the presented material does not provide enough evidence to support the main conclusions namely that the results indicate that the ANN approach outperforms the operational NASA full physics algorithm and that it can be used to improve our knowledge of CO2 fluxes. It is understandable to limit the ANN development to simple cases at first (e.g., nadir only) and it is also not to be expected that the ANN will produce perfect results from the first try on. However, the functionality must be provable. This means it must become clear that the ANN has indeed learned and generalized primarily from the spectral information so that it is able to follow also un-expected CO2 features such as plumes. The presented material seems not sufficient to prove or disprove this. I'm sorry that I cannot give a more positive feedback, but because of this and the many open questions I cannot recommend a publication at AMT. However, due to the relevance of the topic, I encourage the authors to continue their work and, if the results allow it, to resubmit a revised manuscript.*

We have added an analysis (first part of the discussion section) to demonstrate that the Neural Network retrieves the surface pressure and XCO2 from the spectral information and not by simply reproducing the training material.
Conversely, we do not follow the reviewer suggestion to analyze fine scale structures such as plume. First, our first NN attempt only uses a single FOV rather than the 8 cross-track FOV, so that we lack the imagery capabilities that are most useful to identify plumes. Second, this would be a full additional study and is clearly out of the scope of this first paper on the subject.

*General comments: I have some concerns about the suitability of the input parameters to the ANN. As described in my specific comment L86-L88, potentially important information is rejected from the spectra.*
See our answer to the specific comment

*The training data set spans over the same four-year (2015-2018) time period as the test data set and the authors emphasize that this allows XCO2 variations of about 2%. If it is important, that the training covers all possible CO2 concentrations, it is questionable, if a representative training data set can be found suitable also to analyze future OCO-2 data because CO2 is continuously increasing from year to year. What happens when applying the ANN to data from April 2020 including unprecedented largeCO2 concentrations due to the continuous year-to-year increase of atmospheric CO2? Would the ANN still give reasonable results when training*

*only with data from 2015? ANNs are more or less black boxes in the sense, that it is not easily possible to find out which physical relationships they have learned.*

The NN approach must use a training dataset that is representative of the observations to which it is applied. Thus, the NN that we have trained with 2015-2018 data would generate poor results when applied to 2020 observations. We acknowledge that this is a limitation of the approach. However, there has not been real time application of OCO-2 observations so far, so that we do not see that as a strong limitation.

We certainly agree that the approach is a black-box so that its capabilities can only be judged empirically. This is what we attempt in the paper.

*As CO2 is well mixed and long-lived, it is easy to make a relative good guess of its concentration without using any measurements. As an example, by estimating XCO2 only from latitude and time one can achieve already a good agreement with TCCON.*

*Nevertheless, nothing new can be learnt about CO2 from such estimates. Therefore, it is crucial to prove that the ANN's XCO2 is primarily coming from the absorption depth within the spectra. This, however, has not been done.*

We certainly agree that a good a priori can be obtained from the latitude and time. We are well aware of this fact and this is why we do not provide any information on the location and time as input to the ANN. We agree that some indirect information about the latitude and day-in-the-year may be extracted from the observation geometry. Conversely, there is no such indirect information on the longitude and the year of observation (i.e. the viewing geometry is the same from one year to the next, and for all longitudes at a given time). Thus, the ability of the NN to accurately retrieve the ≈2 ppm increment from year to year as well as the zonal gradients is necessarily derived from the spectra and not from the other input data.

*One could confront the ANN with simulated spectra and show that it is indeed able to follow the simulated variations of the CO2. Such simulations could also be used to derive an estimate for the column averaging kernels, e.g., for different solar zenith angles. Additionally, one could apply the ANN to small scenes (with basically constant observation geometry) from which it is known that they include isolated CO2 plumes (see Nassar et al. (2017), or Reuter et al. (2019) for examples). Than it could be analyzed in how far the ANN is capable to follow theXCO2 enhancements and in how far the results agree with those of ACOS.*

We have added a section to demonstrate that the NN does not only mimic the training data and provides additional information. Conversely, we have not followed the reviewer suggestion to analyze XCO2 plumes, as discussed above.

*There are indications for over fitting: as the authors state, the maps in Fig.5 show biases in the test data set which do not exist for the training data set.*

There is little doubt that there is over fitting of the training dataset and this is why we pay attention to have independent training and evaluation data.

*Conventional full physics algorithms allow (and usually require) post filtering by analyzing, e.g., the spectral fit quality but also quantities such as the posteriori XCO2 error estimate. Is this also possible for an ANN approach?*

No. The NN approach is a black box and it does not allow, to our knowledge, a posterior error estimate

*The TCCON validation seems to be not state of the art. For example, it does not consider the averaging kernels. For the ANN this is because they have not been computed, but for the TCCON and ACOS they are available. Which metrics have been used to quantify the*

*performance? Usually, the average single sounding precision and the station-to-station biases are computed as a minimum set of parameters describing the quality. Please note that a low scatter could potentially also be observed when simply ignoring the satellite data which always add instrumental noise. This should also be considered/discussed when comparing CAMS (including no instrumental noise) with TCCON. Please specify what is meant with the different measures of agreement that are used throughout the manuscript (trueness, accuracy, precision, skill, quality).*

We agree that we do not use the averaging kernel as this information is not available for the NN. We have attempted to improve the manuscript and make clear the various comments of the reviewer stated here

Specific comments:

*L19: Please define what exactly is meant with precision (at least within the main text). Is it, e.g., the standard deviation of the retrieved values or the standard deviation of the difference to a truth?*

We feel that there is a clear definition to precision that refers to the standard deviation to some other measure. Here, we do not want to use the term accuracy as we recognize there may be some bias in the validation dataset that we use. We have made a few changes in the text to remove ambiguity

*L42-L44: "Similarly..." This paragraph reads like a description of the light path proxy method, e.g., used by Schneising et al. (2008). However, the idea of full physics algorithms is fundamentally different to this. Because of wavelength dependencies of the surface reflection and of the scattering properties (e.g., optical thickness), the light path is different in the O2 and CO2 bands. Therefore, full physics algorithms use theO2 band to infer knowledge on the scattering particles or processes which allow to estimate the light path in the CO2 band from measurements in the O2 and CO2 band (see, e.g., Butz et al. (2011), Cogan et al. (2012), O'Dell et al. (2018), Reuter et al.(2017), Yoshida et al. (2013)). However, it is correct, that some (not all) full physics algorithms also retrieve the surface pressure.*

We certainly agree with this description of the XCO2 retrieval algorithm. However, our text only gives a rough introduction and provides a reference for a more detailed description for interested readers. We do not feel there is anything misleading in the two sentences

*L45: Molecular (Rayleigh) scattering is not a main difficulty as it is well known. The main problems are aerosols and clouds (see publications cited in my last comment).*

We agree. The use of "molecules" was inappropriate in the sentence and has been changed to "particles"

*L49: Please describe what is meant with "have been optimized".*

We meant "estimated for a best fit between the measured and modeled spectra". The sentence has been changed to "The radiative transfer models that are used for the retrieval leave significant residuals between the measured and modelled spectra, even after the XCO2 and aerosol amount have been inverted for a best fit"

*L53: Why do you consider the signal to be the deviation from the prior? Shouldn't the signal be rather the actual variability.*
The deviation from the prior is the innovation, i.e. the information that is really brought by the observation. If the variability is already known (as for the growth rate or even a large fraction of the seasonal cycle), it cannot be considered as a signal brought by the observation.

*L65-L67: RT models can simulate the radiance usually extremely accurate. However, the input to the RT models (e.g., unknown scattering phase functions, surface BRDF) and approximations needed to meet the requirements on the computational efficiency are the problems. Additionally, there may be unknown instrumental effects (uncertainties in the instrumental line shape function, polarization sensitivity, stray light, etc.).*
Agreed. We added "In addition, there may be some wrong assumptions and unknown instrumental defect that are not accounted for in the forward modeling."

*L71: "The evaluation results show..." should be moved to the discussion.*
Agreed (done)

*L76: Why footprint #4 not #7 or #3? Do the results critically depend on the used footprint?*
There is no reason to select one rather than the others. We have not analyzed whether the results depend on the footprint but see no reason why it would.

*L77: Please discuss if this issue principally will render ANN approaches impossible for glint observations and if not, outline potential solutions.*
The text clearly says that the doppler effect that is significant for the glint -but not so for nadir-observation introduce a complication. Whether this makes the ANN approach impossible for the glint observation cannot be answered before it is attempted. Since then, we have had excellent results with the glint observation, but this is not in the scope of the present paper.

*L80: Have you attempted to remove/mask the most affected pixels?*
Yes. We have removed the pixels on the edge of the spectra, as mentioned in the text

*Sec.2: Please describe which OCO-2 data exactly has been used. Which version and where can be obtained from?*
Quality flags and XCO2 estimates are from Lite V9 whereas the spectra are obtained from product L1B v8. This information is added to the manuscript

*L86-L88: Dividing by the maximum is potentially not ideal because this maximizes the influence of instrumental noise or outliers due to cosmic rays and, additionally, it does not account for slopes in the spectrum. Such effects could be reduced by, e.g., dividing by the 90% percentile of the, e.g., 100 left-most spectral points.*

We agree that the results presented in the paper are for a first attempt that is open to improvements. In practice, we do not divide by the maximum, as incorrectly mentioned in the text, but by something that is similar to the reviewer suggestion: The normalization is based on the mean radiance of the 90-95 percentile range. This is now properly mentioned in the text.

*However, my main concern here is another: Dividing by the maximum radiance removes important information from the spectrum. Namely the information on albedo (as mentioned in*

*the manuscript). As discussed in the literature (see provided references of full physics algorithms), unknown scattering properties introduce among the largest uncertainties in XCO2 retrievals. Knowledge of the albedo is important to infer knowledge of scattering properties. Consider an atmosphere with a surface pressure of 1000 hPa and a scattering layer at 500hPa, reflecting 1% or the incoming radiation. Let CO2 absorb 80% of the radiance along the light path (sun-surface-satellite). This means about 40% will be absorbed along the light path sun-scattering layer-satellite. In the case of an albedo of 100%, the average absorption would be only slightly less than 80%. In the case of an albedo of 0%, the average absorption would amount 40%. This means, the relative depth is not a good measure for the number of particles in the total column. If you would normalize by the solar incoming radiance instead of the maximum radiance you would retain information on albedo, and therefore, also on scattering. Additionally, it shall be noted that, the light path in the O2 band can significantly differ from that in the CO2 band because of differences in the albedo and scattering properties.*

Although we agree with the reviewer comments, NOT dividing by the albedo also causes issues. We felt that it would help the NN training to use an input radiances that vary little outside of the absorption band. Our results indicate that fairly accurate results are obtained with such choice. We certainly agree that the NN approach can be further improved in the future, by us or others. Still, the results obtained with our configuration are, we feel, sufficiently interesting and novel to deserve publication

*L91: The influence of the azimuth should be discussed in the results section.*
We felt it fits better as it is. We have not moved this very short discussion

*L95: 2557 input neurons are quite a lot and results in a rather complex ANN. Often one tries to reduce the dimensionality of the input data by performing, e.g., a PCA. This can probably also help the ANN to generalize instead of memorize. Why have you decided to use the full dimensionality of the input?*
There is no indication that the PCA does a better job than the ANN itself to retain all the information that is available. Use the full dimensionality of the input, which retain all potential information, seems like a natural choice to us

*L96: Why 500 hidden neurons? Is there a rule of thumbs to select a suitable number of hidden neurons? Please discuss how your results depend on the complexity of the network topology. Do you use a so called bias neuron?*
We made a few attempts with a different number of neurons. With 50 neurons, the results were clearly of lower quality. With a larger number of neurons, the training time became significantly larger. The number of neurons could be optimized in the future. We added two sentences in the manuscript.

*L100: Preventing the ANN from over training is certainly important. However, I'm not sure if it is a good idea to stop iterating before convergence is reached. Could the fact that overtraining happens for more than 100 iterations hint at a too complex network topology. Would it be an option to prevent from over fitting by choosing a less complex network with fewer hidden neurons? Additionally, a plot showing the convergence behavior would be nice to have (e.g., RMSD performance of the training and test data vs. number of iterations).*
Note that there is never a full convergence of the network so that a subjective choice of iteration stop is necessary. We do make such convergence plots (see below). We did not feel it provide

significant information. To follow the reviewer suggestion, the convergence plot was added to the appendix

*L103-L104: It is an important point whether the ANN uses information of time and position of the observation or not. Therefore, please discuss, in how far the observation geometry can provide the ANN indirectly with information on the position and/or time of the observations.*
The observation geometry varies with the latitude and the season so that the NN may infer some location information from this input. Conversely, it is the same for one year to the next and, at a given date, for all longitudes. Thus, there is no information on the longitude or the year of observation in the geometry parameters that are provided to the network.

*Which parameters do you mean with "observation geometry" (is, e.g., surface elevation included)?*
The observation geometry is the sun zenith angle and the relative azimuth. The surface elevation is not included

*L104-L106: CO2 does not only have a seasonal variation, but it is also continuously increasing from year to year. Therefore, when having in mind a potential application to future data, the ANN will usually have to deal with concentrations larger than used for the training. How would the ANN behave in such cases? This could, e.g., be answered by confronting the ANN with RT simulated radiances.*
We certainly agree that the NN approach cannot be used to process observations that have been obtained later (or earlier) in time than the training dataset. This point is added in the discussion

*L113: Why do you use ppm as unit for surface pressure?*
Typo corrected. Thanks

*L123: As mentioned in L121, surface based measurements have been used.*
This was a mistake. We meant that no TCCON (surface based remote sensing) observations have been used. It is corrected. Thanks

*L125: Model pressure usually includes water vapor. However, XCO2 is the dry-air column-average. Therefore, strictly speaking, you would have to compute the weights according to the pressure difference corrected for the water vapor content.*
Although it was not explicit in the manuscript, all pressures in the atmospheric transport model that our CAMS product uses (LMDz) are dry air pressures. Thus, there is no need for a correction for the water vapor. We have made it clear in the revised version.

*L128-L137: Interpolating only within the sorted spectrum is a good idea. However, the surface reflectance can introduce significant slopes within the spectra which may significantly change the rank of the spectral pixels. How large is the impact of this effect?*
We have not attempted to quantify this effect

*L166-L171, Fig.3: Why do you use for Fig.3 (left) only soundings where an ACOS retrieval has been made? This drastically reduces the number of cloudy soundings. From Fig.3 (left), I would estimate, that the ANN is capable to derive the surface pressure for definitively cloudy cases nearly bias free with a standard deviation of better than 5hPa.In cases of clouds that are not optically extremely thin, the spectra should not include significant information on the surface pressure. Additionally, the light path is shortened which can usually be interpreted as low*

*surface pressure. Please discuss why the ANN is still capable to derive the surface pressure so well.*

We have acknowledged that our current version of the NN is not independent from the ACOS product. In particular, it uses the results from the Cloud detection algorithm. Only the observation that have been through this detection are used. The "Definitely Cloudy" soundings that are used here have been declared "clear" by the first cloud detection process. There is little doubt that the results would not be the same for the observations that are truly cloudy. Note also that the "cloudy" observations in Figure 3 are rare.

We have added a clarification in the text : "Note also that the observations used hare have all been classified as "clear" by the ACOS pre-processing. Thus, most OCO-2 observations are not used here and Figure 3 should not be interpreted as the ability to retrieve the surface pressure in cloudy conditions"

*L175: I would suggest to also show the corresponding figure for XCO2. If it turns out that the ANN is also capable, to derive XCO2 in definitively cloud contaminated scenes, I would also suggest to add a discussion where this information is coming from.*

See comment above that the "cloudy" observations here is only a very small fraction -probably very specific- of the cloudy observations

*L181-L190: How have you accounted for the column averaging kernels? At least for ACOS and TCCON, this information is available and should be used.*

We have not used the averaging kernels.

*Additionally, the usage of TCCON data should be described in Sec.2 and it should be mentioned in the main text, where the TCCON data can be obtained from and when they were downloaded.*

This information was added in the legend of Table 1

*L206-L210: I have some concerns with this arguing. If you chose a very complex network topology, the ANN might be well able to reproduce CAMS (including systematic persistent biases). If the ANN is too simple, it may not be able to follow the actual CO2 variability. I have the impression that this paragraph implicitly assumes that the network complexity is "just right" so that the ANN was not able to learn biases of CAMS but still generalized enough to follow the actual CO2.*

We certainly do not claim that the NN complexity is "just right". We only give the results for the configuration that we have chosen. As for the argument, it seems clear that if the difference between CAMS and the NN have a random structure, they cannot be used to improve the flux estimates. If they have a spatio-temporal structure, then there is some hope towards that objective.

Note also the discussion in the next paragraph that the structures in the surface pressure are a bad signal toward that objective

*L214: Please discuss why you observe a more or less persistent bias pattern in the even months but not significant differences in the odd months (used for the training). The fact that the training data set performs significantly better than the test data set usually hints at over fitting.*

There is no doubt that there is over-fitting of the data that are used for the training. This is inevitable and this is the reason why we attempt to clearly distinguish the training and evaluation dataset.

*L215: In L206-L210 you suggest that XCO2 difference may come from model deficiencies. Why do you interpret the surface pressure differences as ANN biases?*

Although we do expect biases in the XCO2 modeling (because our knowledge of the surface fluxes is far from perfect), we do not expect such biases in the numerical weather modeling of the surface pressure. Numerous studies have shown that the surface pressure accuracy is better than 1 hPa.

*Fig.5, 7, A1, A2: Please also show the even month used for the training because significant differences between even and odd months, can hint at a potential over fitting.*
There is over-fitting of the training dataset. This is inherent to the method. As a consequence, the even month differences are very small and provide no relevant information

*The shown differences are in the order of 0.5%. What is the expected impact of neglecting the averaging kernels?*
Unfortunately, we do not have this information

*L237: The presented material does not allow this conclusion. Please, particularly, see my specific comments related to the validation method, the used input data for the ANN, and the lack of a prove that the ANN's XCO2 variability is indeed primarily coming from the spectral information.*
We have added some material to show that the information comes primarily from the spectral information. We hope this material is sufficient to convince the reviewer that, indeed, we have enough information for the conclusion that the NN approach allows a high precision. Also note the use of the verb "indicate" and not "demonstrate"

*L238-L239: Please define precision. For which product there is no independent truth (surface pressure or XCO2)?*
This comment applies to both, at the scale of the FOV footprint. The relative accuracy of the "truth" surface pressure (numerical weather mode) is certainly better than that of the XCO2. We have changed the sentences to
However, there are indications that the accuracy on the surface pressure is better than 3 hPa RMS, while the precision (StdDev) of XCO2 is better than 0.8 ppm. Indeed, the data used for the product evaluation has its own error that is difficult to disentangle from that of the estimate based on the satellite observation.

*L260-L263: I would have concerns with both options: i) If the OCO-2 spectra do not include information on, e.g., the upper most CO2, the ANN's AKs will have no sensitivity here independently from the used training data. ii) AKs usually differ from L2 algorithm to algorithm. I would suggest to compute typical AKs by confronting the ANN with simulated spectra for different observation geometries.*
We have added a sentence as per the reviewer suggestion. The various options must be analysed and tested, but it is clearly out of the scope of the present paper. We agree the reviewer suggestion has potential, but may lead to significant computer requirement.

**Technical corrections:**
*L11,43: column integrated CO2 dry air mole fraction -> column-average dry-air mole fraction of CO2*
*L13: uses a full -> is a so called full*
*L38: During -> Along*
*L63: Comprehensive -> Representative*
*L124: observations -> observation*

Done (all of the above)

*Fig1: Please add a legend for the dashed/dotted yellow and red lines. The visibility of these lines is poor. The caption or the axis should include the information which models have been used? r^2 -> r2*
Corrected

*L182, L184: TCCON network -> TCCON*
*L183: Fourier Transform Infrared -> Fourier transform infrared*
*L183: tuned against -> calibrated with*
*L184: "tuning" do you mean "bias correction"?*
Corrected (all of the above)

*L186: "neither...nor" do you mean "either...or"?*
No. we mean the target mode that is neither nadir nor glint (and that is difficult to handle with the NN approach)

*Fig.3: Please increase the font size.*

*Fig.4: Please increase the font size. TCCON station names should start with uppercase letters.*

*Fig.5, 7, A1, A2: The font size is too small, white is ambiguous (snow/ice or delta_p =0), green is not explained.*
All corrected

**References:**
Butz et al. (2011): Butz, A., Guerlet, S., Hasekamp, O., Schepers, D., Galli, A.,Aben,I., Frankenberg, C., Hartmann, J.-M., Tran, H., Kuze,A., Keppel-Aleks, G., Toon, G.,Wunch, D., Wennberg, P.,Deutscher, N., Griffith, D., Macatangay, R., Messerschmidt,J.,Notholt, J., and Warneke, T.: Toward accurate CO2and CH4observations fromGOSAT, Geophys. Res. Lett., 38, L14812,https://doi.org/10.1029/2011GL047888,2011C9

Cogan et al. (2012): Cogan, A. J., Boesch, H., Parker, R. J., Feng, L., Palmer,P. I.,Blavier, J.-F. L., Deutscher, N. M., Macatangay, R., Notholt, J.,Roehl, C.,Warneke, T., and Wunsch, D.: Atmospheric carbondioxide retrieved from the Green-house gases Observing SATel-lite (GOSAT): Comparison with ground-based TCCONobser-vations and GEOS-Chem model calculations, J. Geophys. Res.,117, D21301,https://doi.org/10.1029/2012JD018087, 2012

O'Dell et al. (2018): O'Dell, C. W., Eldering, A., Wennberg, P. O., Crisp, D., Gunson,M.R., Fisher, B., Frankenberg, C., Kiel, M., Lindqvist, H., Man-drake, L., Merrelli, A., Na-traj, V., Nelson, R. R., Osterman, G. B.,Payne, V. H., Taylor, T. E., Wunch, D., Drouin,B. J., Oyafuso,F., Chang, A., McDuffie, J., Smyth, M., Baker, D. F., Basu, S.,Chevallier,F., Crowell, S. M. R., Feng, L., Palmer, P. I., Dubey,M., García, O. E., Griffith, D. W. T.,Hase, F., Iraci, L. T., Kivi,R., Morino, I., Notholt, J., Ohyama, H., Petri, C., Roehl, C.M.,Sha, M. K., Strong, K., Sussmann, R., Te, Y., Uchino, O., and Ve-lazco, V. A.: Im-proved retrievals of carbon dioxide from OrbitingCarbon Observatory-2 with the version8 ACOS algorithm, At-mos. Meas. Tech., 11, 6539–6576, https://doi.org/10.5194/amt-11-6539-2018, 2018

Nassar et al. (2017): Nassar, R., Hill, T. G., McLinden, C. A., Wunch, D., Jones, D.,and Crisp, D.: Quantifying CO2 emissions from individual power plants from space,Geophys. Res. Lett., 44, 10045–10053, https://doi.org/10.1002/2017GL074702, 2017

Reuter et al. (2017): Reuter, M., Buchwitz, M., Schneising, O., Noël, S., Rozanov,V.,Bovensmann, H., and Burrows, J. P.: A Fast AtmosphericTrace Gas Retrievalfor Hyperspectral Instruments Approximat-ing Multiple Scattering – Part 1: Ra-diative Transfer and a Po-tential OCO-2 XCO2Retrieval Setup, Remote Sens., 9,1159,https://doi.org/10.3390/rs9111159, 2017

Reuter et al. (2019): M. Reuter, M. Buchwitz, O. Schneising, S. Krautwurst, C.W.O'Dell, A. Richter, H. Bovensmann, and J.P. Burrows: Computation and analysis of at-mospheric carbon dioxide annual mean growth rates from satellite observations during 2003-2016, Atmos. Chem. Phys., https://www.atmos-chem-phys.net/19/9371/2019,2019

Schneising et al. (2008): O. Schneising, M. Buchwitz, J. P. Burrows, H. Bovens-mann, M. Reuter, J. Notholt, R. Macatangay, T. Warneke: Three years of greenhousegas column-averaged dry air mole fractions retrieved from satellite - Part 1: Carbondioxide. Atmospheric Chemistry and Physics, doi:10.5194/acp-8-3827-2008, 8, 3827-3853, 2008

Yoshida et al. (2013): Yoshida, Y., Kikuchi, N., Morino, I., Uchino, O., Oshchepkov,S.,Bril, A., Saeki, T., Schutgens, N., Toon, G. C., Wunch, D., Roehl,C. M., Wennberg,P. O., Griffith, D. W. T., Deutscher, N. M.,Warneke, T., Notholt, J., Robinson, J., Sher-lock, V., Connor, B.,Rettinger, M., Sussmann, R., Ahonen, P., Heikkinen, P., Kyrö,E.,Mendonca, J., Strong, K., Hase, F., Dohe, S., and Yokota,T.: Improvement of the re-trieval algorithm for GOSAT SWIRXCO2and XCH4and their validation using TCCONdata, At-mos. Meas. Tech., 6, 1533–1547, https://doi.org/10.5194/amt-6-1533-2013,2013Interactive comment on Atmos. Meas. Tech. Discuss., doi:10.5194/amt-2020-177, 2020.

**Reviewer 2 (Chriss O'Dell)**

*Review of David et al., "XCO2 estimates from the OCO-2 measurements using a neural network approach.", by Chris O'Dell.*

*This work details a fast, artificial neural network (NN) approach to retrieving surface pressure and the column-mean dry air mole fraction of CO2(XCO2) from high-spectral resolution measurements in the near-infrared from the Orbiting Carbon Observatory-2 (OCO-2). Traditionally, the most accurate XCO2 retrievals have been from semi-physical("Full-physics") retrievals. These are typically iterative, and typically include accurate calculations of multiple-scattering from thin layers of clouds and aerosols, which makes them exceedingly slow. They also tend to be subject to importance biases (of order 1ppm) due to forward model errors(such as spectroscopic or instrument calibration). A neural-network approach is extremely appealing because it automatically solves the speed problem (NN's are very fast, likely tens of milliseconds per retrieval, vs. minutes for a typical FP approach), and may solve some of the bias problem as well, because they simply train on the "right answer", and do not have to know the details of spectroscopy or instrument models.*

We fully agree with this summary and comment

**General Comments:**
*This work is the first serious effort to use NN's as applied to the XCO2 retrieval problem, and the author's mostly do a good job. However, there are a number of weaknesses and methodological problems in this work that need to be strengthened before I can recommend publication. I see this is a "major revision", but ultimately I believe this work can and should be published, as I believe (for the reasons above) that NN's hold great promise as applied to the XCO2-from-satellites problem.*

We hope the reviewer can be convinced by our additions and corrections in the paper, together with some of our answers below

*Most significantly, as the first reviewer pointed out, it is difficult to ascertain from the manuscript alone exactly what the NN algorithm has learned. The authors trained it on a model (CAMS), and as a first validation, tested it against the same model. Using alternating months as training vs. testing is helpful, but the model certainly has deficiencies that are persistent longer than a month in certain regions, so testing well vs. that same model simply is not a validation. The only other real validation given is against TCCON, which seemed to perform*

*well but because of TCCON's lack of good global coverage, again it is very hard to tell how well the model performs globally in any real sense.*

We have added a discussion and further analysis (beginning of the discussion section) to answer the reviewer comment

*I am also worried about having to re-train the model every year to deal with the ~2 ppm secular increase, which over a mere 4 years is roughly equal to the entire global variability of XCO2.*

This is definitely a weakness of the NN approach that we do not try to hide. Unfortunately, we see no way to fix this weakness. Note however that most scientific studies achieved with the OCO-2 dataset apply to "old" observations so that the ability to process the data in near real-time does not appear essential. We have added a paragraph in the discussion section on this comment.

*I believe it would make their argument much stronger if they ran their algorithm on a few select powerplant cases where the enhancement in the plume is reasonable well-constrained. This is possible for power plants with good bottom-up emissions estimates, and cases where the wind speed is reasonably well-known. See for instance Nassar et al. (2017) for some sample cases. If the NN doesn't see a plume at all, we know that it hasn't been properly trained; if it does, it will dramatically bolster the arguments in this work. Arguing that just because the NN doesn't have direct access to location or time information does not mean it cannot indirectly learn other relationships that allow it to appear to learn well. This is a hypothesis, not a proof that it has learned what you think it did.*

Although we agree that the analysis over plant plumes is something that must be done eventually, it is clearly out of the scope of the paper that is a first try at using the NN approach. As described in the manuscript, we use a single footprint so that our output dataset does not have the "imagery" information that is most useful for the plume. In addition, the identification of plumes in the observation dataset, the use of an atmospheric transport model, and the analysis of each case would be a full study in itself.

*Related to this, I'm somewhat concerned by training on CAMS and then considering to use the resulting XCO2 to correct CAMS. To show that your method works, you almost need to run a full (and fairly complicated) OSSE where you have a true world, a CAMS-like model world with some CO2 errors (spatially and temporally correlated), train on the latter, and then see if you can recover the former with the NN. I don't see how this is guaranteed to work, honestly. How do you know that you won't somehow reproduce systematic errors in CAMS by using the NN approach? You state in the text that you tacitly assume that CAMS errors are not correlated OCO-2 spectra in given areas and for given months. But because the CAMS errors (likely of order 1 ppm) are of a similar magnitude as the XCO2 signal, it is important to point out that this is merely an assumption, and more extensive validation (or a detailed OSSE study) is necessary to prove it.*

To answer this reviewer concern (that are somewhat similar to those of the first reviewer), we have added a full section in the paper. We hope it can convince the reviewers

*Also, you claim to use the "ACOS cloud flag", which you say has values 0,1,2, or 3, as a way to define both your training and testing data sets. I think you mean "Preprocessing Results/cloud_flag_idp" in the L2Std file. If this is correct, please know that this flag is little used by the community. In fact, I've never heard of anyone using it, actually. It was defined about a decade ago for GOSAT and not really touched since then (I verified this with the author of the code that defines it). It has never been carefully validated and it appears to be extremely restrictive ("co2_ratio_idp" must be between 0.99 and 1.01 to pass, which is extremely*

*restrictive and appears to cutout entire regions of the globe). Further, using outcome_flag=1 is also quite restrictive. Can you please comment on these flags, and why you didn't use the far simpler ACOS xco2_quality_flag, which is widely used by the community and is the generally adopted quality flag to use? In the plot below, I have attempted to show the differences between the two approaches for May 2016. I had to match the L2Std files(v8r) to the Lite files(v8), so there may be some differences to what you used in your work, but the general conclusion is that you miss a great deal of data with the highly restrictive data set you are working with. Thus, because it is so restrictive, it may be a far easier task than what ACOS tries to do, which is get the best error possible for the xco2_quality_flag = 0 dataset, which is roughly 6 times bigger.*

We agree that our description of the cloud flags and quality flags used in our study was unclear. We have attempted to correct that.

In practice, we do use only observations with xco2_quality_flag=0. This was not mentioned in the manuscript which is a clear oversight (corrected)

The selection Outcome_flag=1 is used for the training. We have analyzed the Psurf precision statistics as a function of the outcome flag (Figure 3). Since there are no significant difference, the evaluation data and the maps are based on the data with no restriction on the outcome flag. As for the cloud flag, we indeed refer to the Preprocessing Results/cloud_flag_idp. We were not aware that this flag was not used by the community. For the training, we only use cloud_flag=3 (absolutely clear) which is roughly half of the dataset. The analysis of the result precision as a function of this flag (Figure 3) indicates that the results obtained for cloud flag=2 are not significantly different than those with cloud flag=3. On the other hand, those with cloud flag=0 or 1 are significantly poorer. As a consequence, we have retained the observations with cloud flag of 2 or 3, which is 96% of the dataset. Although the cloud flag is not used in the community (as explained to us by the reviewer) it seems that it has some value (cloud flag =0 or 1 is of lesser quality). In the previous version of the manuscript, we used, for the evaluation, only data with cloud flag =3. As a consequence, the number of soundings is significantly increased (a factor of ≈2) and the statistics are slightly changed.

*Finally, in section 2 please give the sources of ACOS/OCO-2 data you used with more specificity. What specific versions and datasets of OCO-2 did you use? V8r, or just V8? Did you use L2Std files, L2Diafiles, Lite files, etc?*

This information is now provided (L2 Lite V9 and the associated warn level and flags from v8 L2lite, and L1b from v8)

*What you train on is pretty critical. I think you should at the very least show a sounding density map of your training (and testing) set.*

The map requested by the reviewer has been added in the supplementary

*Further, I think you should carefully explain your reasoning on how you choose the filtering. You must at least mention the xco2_quality_flag, and ideally you would retrain (or at least test) using this, if you aren't going to define your own quality flags. If you choose to train using very restrictive (clear-sky conservative) filters, please explain this is more detail. Also, both outcome_flag and warn_level (which you use for filtering) come from the ACOS L2FP product (cloud_flag_idp comes from a fast, preprocessor code, the IMAP-DOASPreprocessor, or IDP). It would be much better if you could avoid this entirely, because currently you are throwing away all the soundings that didn't converge or were skipped by the ACOS team, which relies on all the peculiarities of our specific algorithm. To make a useful NN algorithm, it ultimately must be independent of any full physics algorithm, unless you want to train on soundings that*

*pass some smaller subset of data that includes L2FP quality flags, but test on a more complete set of soundings that doesn't use any L2FP quality flags. But you do not appear to do this.*

We certainly agree that, eventually, it will be necessary to become independent of the ACOS preprocessing. This is mentioned explicitly in the discussion section. Conversely, at this stage, we feel it is desirable to process the exact same dataset as that successfully processed by ACOS. This allows a meaningful comparison of the product precision and accuracy.

We have added a few sentences to explain which flags are used, and why we selected those

**Specific Comments:**

*L20, Abstract: I don't think TCCON is a "sunphotometer". That kind of implies more moderate resolution measurements. How about "reference ground-based spectrometer" or something similar?*

We changed photometer to spectrophotometer. We do not think there is any need to be more specific in the abstract. Most readers will know exactly what we are referring to and those wo do not can get the information in the paper.

*L80: Please clarify "a limited set of spectral elements". Make clear these are the solar (Fraunhofer) lines you're talking about.*

Done

*L85(near there): Do you try to mask deep solar lines as well (to remove their Doppler effects)? Please clarify, with a why or why not.*

We have added the sentence : "Conversely, we do not remove the spectra that are affected by the deep solar lines, and let the NN handle these specific features".

*L90 (near there): Might you include the polarization angle directly to the NN, in addition to / instead of the relative azimuth? That might work even better.*

Thanks. We take this suggestion for a future evolution of the algorithm.

*L96: As the readers of this article are likely not NN experts, please discuss the pros and cons of # of hidden layers vs. # of neurons. Also please discuss how was the number 500 chosen or optimized.*

An earlier version of the NN approach used only 50 hidden neurons and led to less accurate results. A higher number or neurons led to much higher training time. We made a few tests, but there was no attempt for an exhaustive analysis and optimization. This could be done for the future although we are satisfied by the current version. We have added two sentences on this subject

*L113: I think you mean "1 hPa", not "1ppm".*

Yes. The typo is corrected

*L125: XCO2 is defined as weighted by the number of dry air molecules per square meter in each layer, not the pressure width. This can be shown to be roughly proportional to dP \* (1-q) for a given layer (e.g., O'Dell et al., 2012), where dP is the pressure width and q is the specific humidity in kg/kg. Please recalculate your model XCO2 using this more standard formulation, if possible, or defend your non-traditional XCO2 definition. The differences are generally small (tenths of a ppm), so it is defensible, but if you can be correct, it is best to do so.*

Although it was not explicit in the manuscript, all pressures in the atmospheric transport model that our CAMS product uses (LMDz) are dry air pressures. Thus, there is no need for a correction for the water vapor. We have made it clear in the revised version.

*L138/Figure 1: This is supposedly for the evaluation dataset, but includes N=381k soundings? In section 2, you say the evaluation dataset only includes 155k soundings. So something is wrong –please explain or fix.*

For the training, we use only the highest quality data as indicated by the outcome flag and cloud detection flag. We then apply the trained NN to a larger fraction of data, with no restriction on the outcome flag and a less strict restriction on the cloud flag (2 or 3). The difference between 155k and 381k is entirely explained by these criteria selection.
We certainly agree that it was poorly explained and we have modified the text for better clarity.

*L138-152: Based on Fig 5, there appears to be some problem in the surface pressure retrieval over mountains, specifically a high bias generally in these regions (visible over the Tibetan Plateau and the U.S. Rocky Mountains). Please discuss.*
In the original version of the paper (second paragraph of section 3), we wrote "…although there is some indication of biases for the lowest pressures that are under-represented in the training dataset." We have added "These biases affect the observations over high elevation surfaces such as the Tibetan Plateau or the US Rocky Mountains"
*I suggest that including the surface elevation in the NN may be a good idea, though technically it shouldn't be necessary.*
Indeed, it does not appear to be necessary, although it would be easy to do so

Regarding TCCON comparison: It would be useful to include the following statistics for ACOS, NN, and CAMS vs. TCCON: Overall Mean, Overall StdDev, and Stddev of Station mean biases. These are useful to evaluate accurate vs. TCCON in simple statistics. See for example Fig 18b in O'Dell et al (2018). It shows a mean bias of Nadir Land observations vs. TCCON of 0.30ppm, and a stddev of 1.04 ppm (it has not calculated the stddev of the station-level mean biases; some groups do this, others not). Finally, it doesn't look like you're applying the averaging kernel (AK) correction when comparing ACOS to TCCON. This typically makes the stddev about 0.1 ppm better. If you do not make this correction, please point this out in the text.
We have added the overall statistics in the text, and the station by station statistics in Table 1. Indeed, we do not use the AK for the validation. We make that clear in the revised version of the manuscript.

*Fig4: Please include a horizontal dashed line so we can see the zero-level. Also, please be clear in the caption or the text if the ACOS and NN are sounding-matched. Typically, when we compare ACOS to TCCON, which use all xco2_quality_flag=0 data. If you were to do this, it may change your results for ACOS vs. TCCON (though better vs. worse, I'm not sure).*
We have added the line as suggested in the manuscript. We only use data with a quality flag of zero at all step of the study. This is now made clear in the manuscript.

***Technical/Grammatical:***

*L92: "Although, the NN technique"à"Although the NN technique"*
*L124: "For each OCO-2 observation"*
*L129: "cosmic flux anomaly"à"cosmic ray flux anomaly"*
*L151: "lowest pressure" "lowest pressures"*
*L181: please replace "classical" with "standard" or "traditional"*
*L250: "that are described in this paper."*
Done (thanks !)